# Object-Centric Refinement for Enhanced Zero-Shot Segmentation

**Srinivasa Rao Nandam**[1]    **Sara Atito**[1,2]    **Zhenhua Feng**[3]    **Josef Kittler**[2]    **Muhammad Awais**[1,2]

[1]Surrey Institute for People-Centred AI (PAI), University of Surrey, UK   [2]CVSSP, University of Surrey, UK
[3]School of AI and Computer Science, Jiangnan University, China

{s.nandam, sara.atito, j.kittler, muhammad.awais}@surrey.ac.uk
fengzhenhua@jiangnan.edu.cn

## Abstract

Zero-shot semantic segmentation aims to recognize, pixel-wise, unseen categories without annotated masks, typically by leveraging vision-language models such as CLIP. However, the patch representations obtained by the CLIP's vision encoder lack object-centric structure, making it difficult to localize coherent semantic regions. This hinders the performance of the segmentation decoder, especially for unseen categories. To mitigate this issue, we propose object-centric zero-shot segmentation (OC-ZSS) that enhances patch representations using object-level information. To extract object features for patch refinement, we introduce *self-supervision-guided object prompts* into the encoder. These prompts attend to coarse object regions using attention masks derived from unsupervised clustering of features from a pretrained self-supervised (SSL) model. Although these prompts offer a structured initialization of the object-level context, the extracted features remain coarse due to the unsupervised nature of clustering. To further refine the object features and effectively enrich patch representations, we develop a dual-stage *Object Refinement Attention (ORA)* module that iteratively updates both object and patch features through cross-attention. Last, to make the refinement more robust and sensitive to objects of varying spatial scales, we incorporate a *lightweight granular attention mechanism* that operates over multiple receptive fields. OC-ZSS achieves state-of-the-art performance on standard zero-shot segmentation benchmarks across inductive, transductive, and cross-domain settings. Code is available at https://github.com/confupload/OC-ZSS.

## 1 Introduction

Semantic segmentation aims to assign a semantic label to every pixel in an image (Xie et al., 2021; Guo et al., 2022). While current methods perform well in closed-world settings, they rely on large-scale pixel-level annotations and struggle to generalize to unseen categories. Zero-shot segmentation (ZSS) addresses this limitation by training on seen classes and leveraging vision-language models such as CLIP (Radford et al., 2021) to generalize to unseen ones. Although CLIP enables zero-shot classification by aligning global image and text embeddings, it lacks mechanisms for fine-grained alignment between visual regions and textual concepts, an essential capability for segmentation.

Recent ZSS methods (Zhou et al., 2023; 2022a; Li et al., 2024; Zhang et al., 2024; Ding et al., 2022) build on frozen CLIP backbones and focus on decoder-side improvements, including cross-attention modules (Zhou et al., 2023), cascading decoders (Li et al., 2024), and sinkhorn-based attention (Kim et al., 2024b). However, these approaches overlook a fundamental limitation: CLIP's patch-level features lack object-centric structure and are less suitable to be naturally grouped into semantically coherent regions (Jiao et al., 2023). This weak visual grounding impairs fine-grained localization and limits segmentation performance, a limitation largely unaddressed by prior ZSS methods.

We argue that refining patch features to be more object-centric, without retraining the encoder, is beneficial for improving zero-shot segmentation. Methods that mine object features have shown promise in supervised (Xu et al., 2022a; Locatello et al., 2020; Zhang et al., 2023) and open-vocabulary settings (e.g., OVSegmentor (Xu et al., 2023)), typically using Slot Attention to produce slot/group

tokens (Locatello et al., 2020). However, in the ZSS setting, the goal is not to generate group tokens but to align patches with text-based semantic classes, rendering these techniques not directly applicable. While CLIP-RC (Zhang et al., 2024) introduces region prompts over fixed square grids, it does not encourage object grouping and does not influence patch-level representations beyond simple feature fusion.

To address this, we propose Object-Centric Zero Shot Segmentor (OC-ZSS), an object-centric zero-shot segmentor that refines patch representations using dynamically discovered object features. A central challenge is identifying object locations without supervision. We introduce a novel mechanism called self-supervision-guided object prompts: frozen prompts injected into the CLIP encoder that attend to distinct object locations via masked attention. These masks are generated by clustering features from a pretrained DINO model (Caron et al., 2021), allowing the prompts to capture coarse object features without requiring annotations or encoder finetuning. Unlike CLIP-RC (Zhang et al., 2024), which applies fixed region prompts, our approach adaptively captures semantically meaningful regions. Additionally, rather than open-vocabulary methods such as CLIP-DINOiser (Wysoczańska et al., 2024b) and ProxyCLIP (Lan et al., 2024), we do not mimic DINO features; DINO is used only to guide the attention of object prompts, which capture object features for refining CLIP patch tokens into more object-centric representations.

These object prompts provide an initial but coarse representation of objects. Meanwhile, CLIP patch features remain object-agnostic. To improve patch features using object cues, we introduce a dual-stage Object Refinement Attention (ORA) module that iteratively refines object and patch features via cross-attention. Unlike standard slot attention (Locatello et al., 2020), which focuses on mining object tokens, ORA enhances patch representations using updated object features while also refining the object features themselves. This mutual refinement promotes tighter semantic grouping and improves alignment with textual concepts. Additionally, in contrast to OV-Segmentor (Xu et al., 2023), which requires end-to-end training of the vision-language stack and generates group tokens, our method operates entirely on frozen CLIP backbones and directly improves patch-text alignment.

Standard cross-attention in ORA focuses on global context aggregation and lacks the ability to capture objects at varying spatial scales, which is crucial for effectively refining both patch and object features. While multi-scale context has traditionally been used in semantic segmentation decoders to improve dense predictions (Chen et al., 2018; Xie et al., 2021; Guo et al., 2022), we incorporate it into the object and patch refinement process itself. Specifically, we propose a granular attention mechanism within ORA that applies depthwise separable atrous convolutions with multiple dilation rates to extract features at different receptive fields. This enables ORA to more effectively model object variability in scale and structure during iterative refinement, improving both patch enrichment and object representation.

Our main contributions are:

- Refine CLIP patch features to be more object-centric, improving segmentation especially of unseen categories.
- Introduce self-supervision-guided object prompts to extract coarse object features from a frozen encoder using unsupervised DINO clustering.
- Design a dual-stage ORA module that jointly refines object and patch features to enhance object-awareness in CLIP representations.
- Incorporate a granular attention mechanism into ORA to capture multi-scale features and improve segmentation robustness.

OC-ZSS achieves state-of-the-art performance on standard zero-shot segmentation benchmarks, including inductive, transductive, and cross-domain settings, highlighting the value of object-centric refinement in vision-language models.

## 2 RELATED WORK

**Zero-Shot Semantic Segmentation:** Zero-shot semantic segmentation (ZS3) aims to segment object categories not seen during training by leveraging external knowledge, typically in the form of class descriptions or text embeddings. Early methods mapped image features and class semantics into a shared space using word embeddings (Gu et al., 2020; Baek et al., 2021; Cheng et al.,

2021). More recent work adopts a transductive setting (Pastore et al., 2021; Bucher et al., 2019), where unseen class names are available during training and pseudo-labels are used to boost performance through self-training. The advent of vision-language models like CLIP (Radford et al., 2021) has enabled more direct pixel-level alignment via text embeddings. Two-stage models such as ZegFormer (Ding et al., 2022) and ZSSeg (Xu et al., 2022b) generate class-agnostic masks before classifying them using text, but incur redundancy due to repeated encoder passes.

FreeSeg (Qin et al., 2023) targets unified/open-vocabulary segmentation with a broader, multi-task recipe based on complex Mask2Former (Cheng et al., 2022) architecture. MAFT (Jiao et al., 2023) fine-tunes the CLIP image encoder with mask-aware and distillation losses to improve proposal sensitivity in dual-stage methods. These heavier multi-stage systems motivated single-stage, parameter-efficient alternatives that align CLIP features and text in single setting. MaskCLIP+(Zhou et al., 2022a) modifies CLIP's penultimate layer to better match textual embeddings. ZegCLIP (Zhou et al., 2023) and CLIP-RC (Zhang et al., 2024) introduce prompts or spatial tokens to adapt CLIP's vision encoder. Cascade-CLIP (Li et al., 2024) proposes hierarchical decoders, while OT-Seg (Kim et al., 2024b) replaces standard attention with Sinkhorn attention. MVP-SEG+(Guo et al., 2023) adds orthogonality constraints to enhance part-level distinction. SPT-SEG (Xu et al., 2024b) introduces spectral prompts with a spectral-guided decoder. Unlike parameter efficient methods described earlier, a different direction is taken by AlignZeg (Ge et al., 2024), which trains an auxiliary vision encoder to reduce the domain gap between image-level and pixel-level features. It also applies prompt learning to adapt the CLIP vision encoder, and uses two decoders: a pixel decoder for class-agnostic mask proposals and a semantic decoder built from CLIP's top layers, fusing these streams end to end. In contrast, we keep CLIP frozen and use a single lightweight decoder, remaining parameter-efficient like prior decoder methods (Kim et al., 2024b; Xu et al., 2024b; Zhang et al., 2024). Rather than pursuing increasingly intricate decoders, we further differentiate by refining patch features through ORA with object-level cues derived from object prompts, which improves object localization and promotes better generalization to unseen classes.

Open-vocabulary semantic segmentation methods often pair CLIP with self-supervised models, primarily DINO (Caron et al., 2021) or DINOv2 (Oquab et al., 2024), to improve token grouping. FOSSIL (Barsellotti et al., 2024) retrieves synthetic visual references and aligns them to masks produced from DINO features using OpenCut. Lazy Visual Grounding builds an affinity graph over DINO tokens and segments it with normalized cuts before assigning text labels (Kang & Cho, 2024). CLIP-DINOiser distills DINOs localization priors into CLIP, teaching CLIP to produce DINO-like local features (Wysoczańska et al., 2024b). Similarly, ProxyCLIP produces DINO-like grouping by forming proxy attention from correspondences computed by DINO, with adaptive normalization and masking (Lan et al., 2024). In contrast, we do not mimic or distill DINO: we introduce object-centric structure into CLIP features, using DINO only to guide object prompts, then use the object features from these prompts within Object Refinement Attention (ORA) to iteratively refine both patch and object features, yielding more generalisable, object-centric CLIP patch representations.

Several other works also explore semantic segmentation in more general open-vocabulary settings, where their training and evaluation protocols differ from the standard inductive/transductive zero-shot segmentation setup. CLIP-DIY (Wysoczańska et al., 2024a) performs training-free open-vocabulary semantic segmentation by applying multi-scale dense CLIP inference guided by unsupervised object localization. SAM-CLIP (Wang et al., 2024) merges SAM and CLIP into a unified vision backbone via multi-task distillation on large-scale web data, and targets open-vocabulary segmentation. Shape-aware ZSS (Liu et al., 2023) promotes shape-aware segmentation by fusing CLIP-based predictions with eigensegments from self-supervised spectral decomposition. SCD (Qiu et al., 2025) fine-tunes CLIP ViTs with a Spatial Correlation Distillation framework to improve region-language alignment for open-vocabulary dense prediction. SimZSS (Stegmüller et al., 2025) aligns a text encoder to a frozen vision-only backbone using imagecaption datasets such as COCO Captions and LAION, and is evaluated under open-vocabulary segmentation protocols. In contrast, OC-ZSS is strictly studied under the inductive/transductive zero-shot segmentation setting with explicit seen/unseen splits, as in standard ZSS methods such as OTSeg and CLIP-RC.

**Object-Centric Representation and Slot Attention** Slot Attention (Locatello et al., 2020) introduced an iterative attention mechanism for learning object-centric representations by binding latent slots to visual inputs. While effective for unsupervised object discovery, it suffers from issues such as slot collapse, addressed in later work using optimal transport-based objectives (Zhang et al., 2023). The framework has also been extended to video segmentation: Guided Slot Attention (Lee et al.,

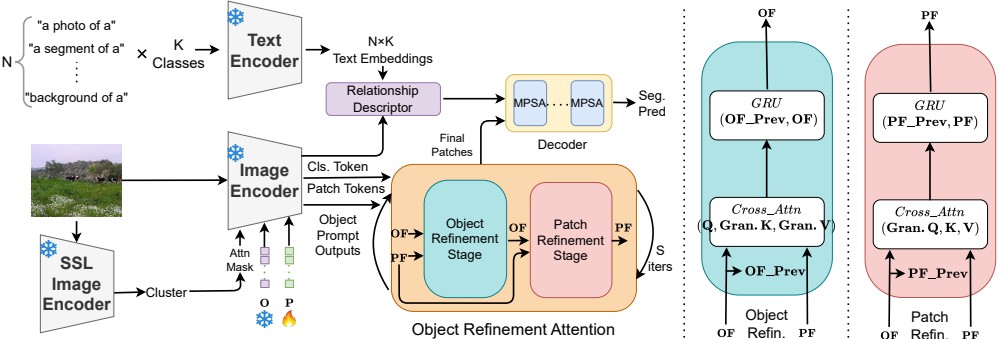

Figure 1: Architecture overview of OC-ZSS. Object prompts ($\mathbf{O}$), guided by an attention mask from a frozen SSL encoder, are passed with learnable prompts ($\mathbf{P}$) and the input image through an image encoder to produce patch ($\mathbf{PF}$) and coarse object features ($\mathbf{OF}$). Patch refinement is performed via a dual-stage ORA module with $S$ iterations. Refined patches are then passed to a decoder for segmentation. Patch/object refinement with granular attention is shown on the right.

2024) enforces temporal consistency by propagating slots across frames, while Adaptive Slot Attention (Fan et al., 2024) adjusts the number of slots dynamically over time. Recent approaches like EAGLE (Kim et al., 2024a) and SOLV (Aydemir et al., 2023) further explore object-centric modeling by discovering spatial or temporal object masks through clustering or reconstruction-based objectives. In the open-vocabulary setting, OVSegmentor (Xu et al., 2023) applies Slot Attention within a trainable vision encoder to mine group tokens but does not refine patch-level features and requires full end-to-end training. In contrast, our method operates on frozen CLIP features, using externally guided object prompts to enhance patch representations. We further introduce multi-scale refinement, a direction not explored in prior slot-based methods.

## 3 METHODOLOGY

We propose **OC-ZSS**, an object-centric framework for zero-shot segmentation that improves CLIP patch representations by modeling and refining object-level structure within a frozen backbone. Existing zero-shot segmentation methods (Zhou et al., 2023; Zhang et al., 2024; Kim et al., 2024b) typically focus on aligning CLIP features to text embeddings via decoder-side attention, but overlook the fact that the underlying patch features lack object-centric grouping. While some methods apply auxiliary supervision like OTSeg (Kim et al., 2024b) or fuse with static region prompts like CLIP-RC (Zhang et al., 2024), they do not explicitly extract or refine object features, limiting their ability to support fine-grained segmentation of unseen classes. To address this, we introduce frozen self-supervision-guided object prompts that are injected into the CLIP encoder that attend to distinct object regions via masked attention. These attention masks are constructed by clustering features from a pretrained DINO model, allowing each prompt to focus on a coarse object region without requiring labeled masks or encoder finetuning.

However, these object prompts provide only coarse object features and do not directly influence the patch features used for segmentation. To refine both representations, we introduce the Object Refinement Attention (ORA) module, which performs iterative cross-attention between object and patch features. ORA gradually enhances object tokens and uses them to enrich patch semantics, promoting tighter object-level grouping and improved alignment with text embeddings. Standard attention layers used in refinement struggle to capture objects with varying spatial extents. To address this, we incorporate a granular attention mechanism into ORA by replacing linear projections with multi-scale feature extractors based on depthwise separable atrous convolutions (Chen et al., 2018). This enables both object and patch features to be refined in a scale-aware manner, improving robustness to object size variation. An overview of OC-ZSS is shown in Figure 1. We describe each component in detail in the following subsections.

## 3.1 PRELIMINARIES

CLIP (Radford et al., 2021) is used as the vision-language backbone, with a prompted vision encoder and a multi-prompt Sinkhorn attention decoder for the task of zero-shot segmentation (Kim et al., 2024b). During training, the model is exposed only to a set of seen classes $C_s$, and is evaluated on both seen and unseen categories $C = C_s \cup C_u$. Class-level text embeddings are obtained from the CLIP text encoder using $K$ prompt templates, yielding $\mathbf{T} = \text{Text\_Enc}(C) \in \mathbb{R}^{K \times N \times D}$, where $N$ is the number of classes and $D$ is the embedding dimension. The CLIP vision encoder produces outputs $\{g, P, H\}$: the global class token $g$, deep prompts $P = \{p_0, \ldots, p_{np}\}$, and patch embeddings $H = \{h_1, \ldots, h_M\}$. Following (Kim et al., 2024b), a relationship descriptor for each class is constructed as $\hat{\mathbf{T}} = \text{concat}(T \odot g, T)$, which serves as the query input to a decoder composed of three Multi-Prompt Sinkhorn Attention (MPSA) blocks, where MPSA uses multi-prompt Sinkhorn (MPS) attention (details in appendix). Each block performs:

$$\text{MPSA}(\mathbf{Q}, \mathbf{K}, \mathbf{V}) = \text{MPS}(\mathbf{Q}\mathbf{K}^T)\mathbf{V}, \tag{1}$$

An auxiliary prediction $\tilde{\mathbf{Y}}$ is also generated directly from the relationship descriptor $\hat{\mathbf{T}}$ before the decoder and on $H$ to better align patches with the final segmentation, where Upsample (bilinear interpolation) restores input resolution and Sigmoid is applied element-wise.

$$\tilde{\mathbf{Y}} = \text{Upsample}\left(\text{Sigmoid}\left(\text{MPS}(\hat{\mathbf{T}}\mathbf{H}^T)\right)\right). \tag{2}$$

The final output during inference is the average of $\mathbf{Y}$ and $\tilde{\mathbf{Y}}$. The total training loss is defined as:

$$\mathcal{L}_{tot} = \mathcal{L}_{seg}(\mathbf{Y}, \mathbf{Y}_{gt}) + \mathcal{L}_{seg}(\tilde{\mathbf{Y}}, \mathbf{Y}_{gt}), \tag{3}$$

where $\mathcal{L}_{seg}$ combines Dice and focal loss, and $\mathbf{Y}_{gt}$ denotes the ground-truth segmentation map.

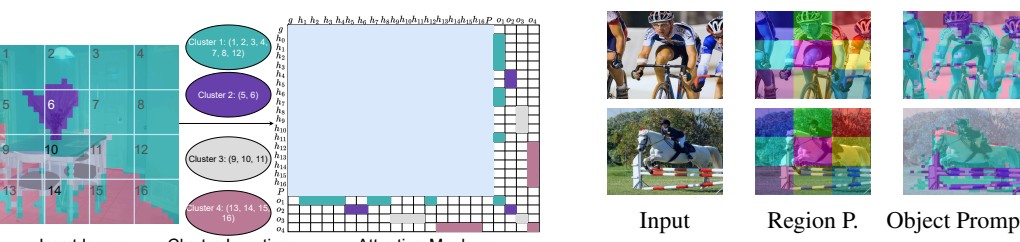

Figure 2: Clustered image and patch locations used to generate the attention mask. Clusters are color-coded in both the image and mask; white indicates $-\infty$, and other colors indicate 0 in the mask.

Figure 3: Region prompts attend to squares; object prompts focus on dynamic objects. Colored regions distinguish areas.

## 3.2 SELF-SUPERVISION-GUIDED OBJECT PROMPTS FOR CAPTURING RICH OBJECT FEATURES

Our framework is built on enhancing patch representations through object-level features. To enable object-based refinement, we first need to capture plausible object features that provide an initial basis for further refinement. At each transformer layer $l$ of the CLIP vision encoder, the input is denoted as $\mathbf{X} = \{\mathbf{g}^l, \mathbf{P}^l, \mathbf{H}^l, \mathbf{O}^l\}$, where $\mathbf{g}^l$ is the class token, $\mathbf{P}^l$ are learnable prompts, $\mathbf{H}^l$ are patch embeddings, and $\mathbf{O}^l = \{\mathbf{O}^l_1, \ldots, \mathbf{O}^l_{no}\}$ are frozen object prompts appended at that layer.

The goal is to guide each object prompt to attend to a distinct, semantically meaningful object location in the image. Prior work such as CLIP-RC (Zhang et al., 2024) has used region prompts to focus on fixed square locations via attention. However, fixed grids cannot accurately capture the non-rigid and often irregular shapes of objects, which is essential for enabling object-centric refinement. Figure. 3 shows difference between attention of region and object prompts.

To overcome this, we generate a custom attention mask Attn_mask $\in \mathbb{R}^{(1+np+no+M) \times (1+np+no+M)}$ that selectively enables each object prompt to focus only on a particular subset of patch tokens. Figure. 2 shows the attention generation process. The mask is initialized to zeros, and rows and

columns corresponding to object prompts are first set to $-\infty$ to block self-attention. To allow each object prompt to attend to its corresponding region, selected entries are set back to zero.

Constructing such masks for square regions is trivial, but capturing objects requires dynamic, non-rigid regions. To this end, we use a pretrained self-supervised model (DINO (Caron et al., 2021)) to generate object-aware patch groupings. The input image $\mathbf{I}$ is first patchified and passed through DINO to obtain patch-level features $\mathbf{X}^s$. These features are then clustered into $no$ groups using Voronoi clustering (Aurenhammer, 1991), where implementation follows (Zhang et al., 2022) (details in suppl.). Let $C = \{c_1, \ldots, c_{no}\}$ be the sets of patch indices assigned to each object prompts.

$$C = \{c_1, \ldots, c_{no}\} = \text{voronoi\_clustering}(\mathbf{X}^s) \tag{4}$$

The attention mask is updated for each object prompt $j$ to allow attention to patch tokens in $c_j$:

$$\text{Attn\_mask}[1 + np + M + j, 1 + np + i] = 0 \quad \forall i \in c_j \tag{5}$$

$$\text{Attn\_mask}[1 + np + i, 1 + np + M + j] = 0 \quad \forall i \in c_j \tag{6}$$

This attention masking strategy is applied across all $L$ self-attention layers in the encoder. As a result, each object prompt $\mathbf{O}_j^l$ learns to focus on its assigned region across layers, gradually capturing object-level information. The final set of object prompts from the last layer, $\mathbf{OF} = \mathbf{O}^L$, serves as coarse object features for our Object Refinement Attention (ORA) module.

### 3.3 OBJECT REFINEMENT ATTENTION MODULE

To refine patch features using the coarse object features extracted from object prompts, we propose a dual-stage Object Refinement Attention (ORA) module, which also refines these object features themselves. This module operates iteratively, with each iteration consisting of two stages: object refinement followed by patch refinement.

In the object refinement stage, the initial object features $\mathbf{OF}$ are taken from the final layer of the CLIP encoder, i.e., $\mathbf{OF} = \mathbf{O}^L$, and refined based on the patch features $\mathbf{PF} = \mathbf{H}^L$. Proper initialization of $\mathbf{OF}$ is critical, as random initialization has been shown to degrade performance in iterative refinement methods such as slot attention (Zhang et al., 2023). In our case, the object prompts introduced into the encoder not only serve to extract object-specific features but also provide semantically meaningful initialization for $\mathbf{OF}$, which is essential for stable refinement. This stage begins with a cross-attention operation where object features attend to patch features, producing refined object features $\mathbf{OF\_Ref}$:

$$\mathbf{OF\_Prev} = \mathbf{OF}; \quad \mathbf{OF\_Ref} = \text{Cross\_Attention\_OR}(\mathbf{OF}, \mathbf{PF}) \tag{7}$$

These features are then dynamically aggregated with the previous object features using a recurrent layer, specifically a Gated Recurrent Unit (GRU) (Cho et al., 2014). This recurrent update is crucial for stabilizing the refinement process over iterations by maintaining continuity in object representations, as shown effective in prior iterative attention frameworks (Locatello et al., 2020).

$$\mathbf{OF} = \text{GRU}(\mathbf{OF\_Prev}, \mathbf{OF\_Ref}) \tag{8}$$

Next, in the patch refinement stage, the updated object features $\mathbf{OF}$ are used to refine the patch features $\mathbf{PF}$. This stage mirrors the object refinement step but inverts the attention direction. Patch features attend to object tokens via a second cross-attention block:

$$\mathbf{PF\_Prev} = \mathbf{PF}; \quad \mathbf{PF\_Ref} = \text{Cross\_Attention\_PR}(\mathbf{PF}, \mathbf{OF}) \tag{9}$$

The refined patch features are again passed through a GRU to yield the updated stable patch representations:

$$\mathbf{PF} = \text{GRU}(\mathbf{PF\_Prev}, \mathbf{PF\_Ref}) \tag{10}$$

These two refinement stages are repeated for $S$ iterations. The final patch features $\mathbf{PF}$ serve as object-aware representations and are passed to the segmentation decoder. Specifically, the key and value matrices $\mathbf{K}, \mathbf{V}$ in the Multi-Prompt Sinkhorn Attention decoder (Equation 1) are derived from $\mathbf{PF}$ instead of the original encoder outputs $\mathbf{H}$. The rest of the decoding and training setup, including loss computation and prediction aggregation, follows OTSeg (Kim et al., 2024b) as described in Section 3.1: Equation 1 through Equation 3.

### 3.4 GRANULAR ATTENTION FOR BETTER OBJECT CAPTURE

In our ORA module, let $\mathbf{K}^{or}, \mathbf{V}^{or}$ be the key and value for the object refinement attention Cross_Attention_OR, and $\mathbf{Q}^{pr}$ be the query for the patch refinement attention Cross_Attention_PR, which are typically derived from patch features $\mathbf{PF}$ using linear projections. However, such projections are limited in their ability to capture context at varying spatial resolutions, an essential requirement for refining object and patch-level representations corresponding to objects of different scales.

To address this, we incorporate a multi-scale feature extractor into ORA that replaces the linear projections to make the attention more granular. Motivated by prior work in semantic segmentation (Xie et al., 2021; Guo et al., 2022; Chen et al., 2018), we adapt a lightweight module inspired by (Chen et al., 2018), but apply it within our object and patch refinement stages rather than in a decoder. This module uses parallel atrous depthwise convolutions (DW_Conv) with dilation rates $d_1, d_2, d_3, d_4$ and a global average pooling (GAP) branch. The outputs are concatenated along channels and passed through a $1 \times 1$ convolution to reduce channels from $5D$ to $D$, where $D$ is the original embedding dimension.

$$\mathbf{X}_{in} = \text{Concat}(\text{DW\_Conv}(3 \times 3, r = d_1), \text{DW\_Conv}(3 \times 3, r = d_2),$$
$$\text{DW\_Conv}(3 \times 3, r = d_3), \text{DW\_Conv}(3 \times 3, r = d_4), \text{GAP}) \quad (11)$$

$$\mathbf{X}_{gran} = \text{Conv}(1 \times 1, 5D, D)(\mathbf{X}in) \quad (12)$$

We apply this module to generate the granular key and value for object refinement, $\mathbf{Gran\_K}^{or}, \mathbf{Gran\_V}^{or} = \text{multi-scale}(\mathbf{PF})$, and the granular query for patch refinement, $\mathbf{Gran\_Q}^{pr} = \text{multi-scale}(\mathbf{PF})$. Here multi-scale layer is application of Equations 11, 12.

By replacing the standard projections in Cross_Attention_OR and Cross_Attention_PR with these scale-aware representations, the ORA module becomes more granular and robust to spatial variation, enabling finer object capture and more discriminative patch features.

## 4 EXPERIMENTS

We evaluate our proposed OC-ZSS framework on three widely used zero-shot semantic segmentation (ZSS) benchmarks: PASCAL VOC 2012 (20 object categories)(Everingham et al., 2010), PASCAL Context (59 classes)(Mottaghi et al., 2014), and COCO-Stuff 164K (171 classes)(Caesar et al., 2018), using standard seen/unseen splits from prior works(Ding et al., 2022; Zhou et al., 2023; Zhang et al., 2024). Performance is measured using mean Intersection-over-Union (mIoU) for both unseen (U) and seen (S) classes, with the harmonic mean (hIoU) to capture the balance between the two. We report results under both the inductive setting where training uses only annotations from seen classes and the transductive setting, which allows access to unlabeled images containing unseen class instances during training (Zhou et al., 2023). OC-ZSS uses CLIP ViT-B/16 (Radford et al., 2021) as the backbone, keeping the text encoder and DINO-B/16 (Caron et al., 2021) SSL encoder frozen, while applying visual prompt tuning (VPT) (Jia et al., 2022) on the image encoder. The decoder is a 3-layer lightweight transformer using Multi-Prompt Sinkhorn Attention (MPSA). We set 6 object prompts for VOC and COCO-Stuff, and 8 for PASCAL Context, with 4 dual-stage refinement iterations by default. The model is trained using AdamW (Loshchilov & Hutter, 2019) with a learning rate of 2.5e-5 and batch size of 16, for 20K, 40K, and 80K iterations on VOC 2012, PASCAL Context, and COCO-Stuff respectively. All experiments are run on a single NVIDIA A100 or four RTX 3090 GPUs.

**Inductive & Transductive Setting.** We evaluate OC-ZSS under both inductive and transductive zero-shot segmentation settings across three widely used benchmarks: PASCAL VOC 2012, PASCAL Context, and COCO-Stuff 164K. In our comparisons with OTSeg (Kim et al., 2024b), we report results using its best-performing variant, OTSeg+. In the *inductive setting*, where the model is trained only with annotations from seen classes and without access to unseen class names, OC-ZSS achieves the best overall performance. As shown in Table 1, OC-ZSS attains 85.3% mIoU for unseen classes and 93.6% for seen classes on VOC 2012, resulting in a harmonic mean of 89.2%, outperforming OTSeg and CLIP-RC. On PASCAL Context, it achieves 62.1% (U) and 55.6% (S), while on COCO-Stuff 164K it reaches 42.8% and 42.2%, both higher than prior methods.

Table 1: Comparison under the inductive setting (* denotes our reproduction, official code).

| Methods | VOC 2012 | | | PASCAL Context | | | COCO-Stuff164K | | |
|---|---|---|---|---|---|---|---|---|---|
| | mIoU(U) | mIoU(S) | hIoU | mIoU(U) | mIoU(S) | hIoU | mIoU(U) | mIoU(S) | hIoU |
| ZegFormer (Ding et al., 2022) | 63.6 | 86.4 | 73.3 | - | - | - | 33.2 | 36.6 | 34.8 |
| ZSSeg (Xu et al., 2022b) | 72.5 | 83.5 | 77.6 | - | - | - | 36.3 | 39.3 | 37.8 |
| MAFT (ZSSeg) (Jiao et al., 2023) | 66.2 | 88.4 | 75.7 | - | - | - | 40.1 | 40.6 | 40.3 |
| ZegCLIP (Zhou et al., 2023) | 77.8 | 91.9 | 84.3 | 54.6 | 46.0 | 49.9 | 41.4 | 40.2 | 40.8 |
| SPT-SEG* (Xu et al., 2024b) | 81.9 | 91.1 | 86.2 | 57.3 | 52.8 | 55.0 | 39.8 | 41.5 | 40.6 |
| CLIP-RC (Zhang et al., 2024) | 80.7 | 91.6 | 85.8 | 57.3 | 47.5 | 51.9 | 41.6 | 40.9 | 41.2 |
| OTSeg (Kim et al., 2024b) | 81.6 | 93.3 | 87.1 | 60.4 | 55.2 | 57.7 | 41.8 | 41.3 | 41.5 |
| OC-ZSS | **85.3** | **93.6** | **89.2** | **62.1** | **55.6** | **58.7** | **42.8** | **42.2** | **42.5** |

Table 2: Comparison under transductive setting (* denotes our reproduction, official code).

| Methods | VOC 2012 | | | PASCAL Context | | | COCO-Stuff164K | | |
|---|---|---|---|---|---|---|---|---|---|
| | mIoU(U) | mIoU(S) | hIoU | mIoU(U) | mIoU(S) | hIoU | mIoU(U) | mIoU(S) | hIoU |
| ZSSeg (Xu et al., 2022b) | 78.1 | 79.2 | 79.3 | - | - | - | 43.6 | 39.6 | 41.5 |
| MaskCLIP+ (Zhou et al., 2022a) | 88.1 | 86.1 | 87.4 | 66.7 | 48.1 | 53.3 | 54.7 | 39.6 | 45.0 |
| FreeSeg (Qin et al., 2023) | 82.6 | 91.8 | 86.9 | - | - | - | 49.1 | 42.2 | 45.3 |
| MVP-SEG+ (Guo et al., 2023) | 87.4 | 89.0 | 88.0 | 67.5 | 48.7 | 54.0 | 55.8 | 39.9 | 45.5 |
| ZegCLIP (Zhou et al., 2023) | 89.9 | 92.3 | 91.1 | 63.2 | 47.2 | 55.6 | 59.9 | 40.7 | 48.5 |
| SPT-SEG* (Xu et al., 2024b) | 93.4 | 93.5 | 93.4 | 68.2 | 53.4 | 59.9 | 60.4 | 42.2 | 49.7 |
| CLIP-RC (Zhang et al., 2024) | 92.2 | 93.9 | 93.0 | 64.5 | 48.1 | 55.1 | 60.8 | 42.0 | 49.7 |
| OTSeg (Kim et al., 2024b) | 94.3 | 94.3 | 94.4 | 67.0 | 54.0 | 59.8 | 62.6 | 41.4 | 49.8 |
| OC-ZSS | **95.6** | **94.9** | **95.2** | **70.0** | **55.6** | **61.9** | **64.2** | **43.0** | **51.5** |

In the *transductive setting*, where the model is trained with access to unseen class names (no unseen class seg. ground truth), OC-ZSS further improves across all benchmarks. It achieves 95.6% and 94.9% mIoU for unseen and seen classes respectively on VOC 2012, with a harmonic mean of 95.2%. On PASCAL Context and COCO-Stuff, it reaches hIoU scores of 61.9% and 51.5%, setting a new benchmark. These consistent improvements can be attributed to our object-centric patch refinement framework. Unlike prior methods, our approach focuses on generating patches that capture entire object regions rather than arbitrary parts. OC-ZSS first identifies coarse object-level features, which are then refined in subsequent stages to form more accurate and object-focused patch groupings. This structured refinement process helps the model better learn object semantics and improves generalization especially in unseen classes, which is important in zero-shot segmentation.

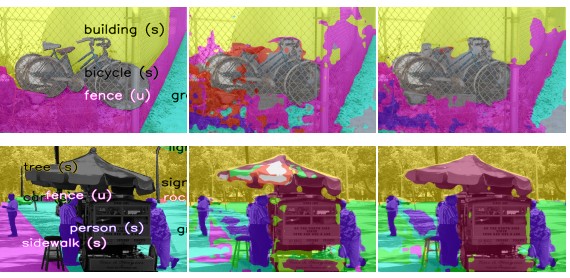
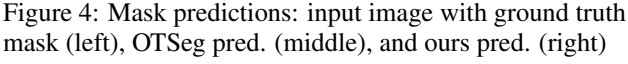
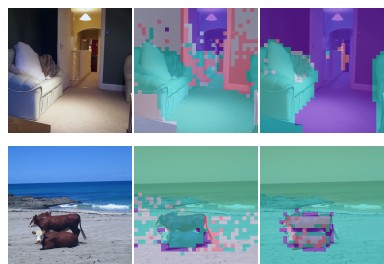

Figure 4: Mask predictions: input image with ground truth mask (left), OTSeg pred. (middle), and ours pred. (right)

Figure 5: Input image (left), clustered patches without ORA (middle), and clustered patches with ORA (right)

Figure 4 presents a comparison between the predictions of the previous state-of-the-art method, OTSeg (Kim et al., 2024b), and our proposed OC-ZSS. The predictions from OTSeg exhibit significant noise and reduced accuracy, whereas our method produces cleaner and more precise segmentation maps across both seen and unseen categories. Figure 5 illustrates the impact of our Object-aware Refinement and Aggregation (ORA) module on patch clustering. We visualize clusters by assigning different colors to patches within an input image. Without the ORA module, the patch clusters appear scattered and unstructured. In contrast, with ORA, the clusters are more coherent and localized around meaningful objects. This demonstrates that object-centric patches not only contribute to improved quantitative performance but also lead to more semantically meaningful and visually accurate segmentation outputs.

Table 3: Cross-dataset generalization using COCO as the source.

| Method | PASCAL Context | VOC 2012 |
|---|---|---|
| **Inductive setting** | | |
| ZegCLIP | 41.2 | 93.4 |
| OTSeg | 49.1 | 93.8 |
| OC-ZSS | **49.6** | **94.2** |
| **Transductive setting** | | |
| ZegCLIP | 45.8 | 94.2 |
| OTSeg | 53.4 | 94.2 |
| OC-ZSS | **54.0** | **94.5** |

Table 4: Model efficiency and resource usage.

| Method | Params (M) ↓ | GFLOPs ↓ | GPU Mem ↓ |
|---|---|---|---|
| Zsseg | 61.1 | 1916.7 | – |
| ZegFormer | 60.3 | 1829.3 | – |
| CLIPRC | 36.9 | 73.8 | 3.47G |
| OTSeg | 13.8 | 61.9 | 3.25G |
| OC-ZSS | 27.2 | 64.0 | 3.35G |

Table 5: VOC12 with very few seen classes

| Methods | VOC 25% | | | VOC 50% | | |
|---|---|---|---|---|---|---|
| | U | S | hIoU | U | S | hIoU |
| CLIP-RC (Zhang et al., 2024) | 28.8 | 71.9 | 41.1 | 55.1 | 86.3 | 67.3 |
| OTSeg (Kim et al., 2024b) | 17.7 | 88.3 | 29.4 | 53.0 | 90.2 | 66.8 |
| OC-ZSS | **49.8** | **92.7** | **64.8** | **59.1** | **90.6** | **71.5** |

Table 6: Ablation of main components for VOC 2012 and Context (inductive setting).✓ represents component used.

| Components | | | VOC 2012 | | | PASCAL Context | | |
|---|---|---|---|---|---|---|---|---|
| ORA | Obj.prompts | Gran. Attn | mIoU(U) | mIoU(S) | hIoU | mIoU(U) | mIoU(S) | hIoU |
| ✗ | ✗ | ✗ | 80.6 | 92.5 | 86.1 | 59.4 | 55.0 | 57.1 |
| ✓ | ✗ | ✗ | 82.0 | 93.0 | 87.1 | 61.4 | 55.1 | 58.0 |
| ✓ | ✓ | ✗ | 84.0 | 93.2 | 88.3 | 61.7 | 55.4 | 58.4 |
| ✓ | ✓ | ✓ | **85.3** | **93.6** | **89.2** | **62.1** | **55.6** | **58.7** |

Table 7: Attention masks generation strategies for object prompts.

| Variant | hIoU |
|---|---|
| DINO-B/16 (Caron et al., 2021) | **89.2** |
| DINO-B/8 (Caron et al., 2021) | 88.4 |
| DINOv2-B/14 (Oquab et al., 2024) | 89.1 |
| CLIP (Radford et al., 2021) | 87.6 |
| Region (Zhang et al., 2024) | 87.2 |
| No Mask | 87.1 |

**Cross-Dataset Generalization.** OC-ZSS demonstrates strong cross-dataset generalization under both inductive and transductive settings. As shown in Table 3, when trained on COCO and evaluated on PASCAL Context and VOC 2012, OC-ZSS outperforms prior methods across both datasets. In the inductive setting, it achieves 49.6% mIoU on PASCAL Context and 94.2% on VOC 2012, surpassing ZegCLIP and OTSeg. Under the transductive setting, performance further improves to 54.0% and 94.5%, respectively. These results validate the generalization ability of our object-centric refinement framework across distribution shifts.

**Maintaining Efficiency with Object-Centric Design.** OC-ZSS introduces additional parameters to support its refinement architecture, yet remains highly efficient compared to other methods. As shown in Table 4, it uses only 3.35G GPU memory and 64.0 GFLOPs, staying close to OTSeg despite delivering significantly better segmentation. The slight increase in parameter count (27.2M) is a deliberate and lightweight addition, enabling object-centric reasoning without relying on extra encoders or complex pipelines. Unlike CLIP-RC, which incurs higher memory and compute costs, OC-ZSS achieves strong segmentation performance with minimal overhead, demonstrating that substantial gains can be achieved without significant increases in complexity.

**Strong Performance in Low-Shot Regimes.** We further evaluate OC-ZSS under labeled data scarcity, where only 25% or 50% of PASCAL VOC classes are treated as seen during training, compared to the default 75% setting (15 seen classes). As shown in Table 5, OC-ZSS achieves 49.8% mIoU on unseen classes with just 25% supervision and 59.1% with 50%, outperforming both CLIP-RC and OTSeg. The strong performance under the 25% setting in particular demonstrates the models ability to generalize effectively from minimal supervision, validating the impact of our patch refinement framework.

## 5 ABLATIONS

**Impact of Each Component in OC-ZSS.** We conduct a stepwise ablation on VOC 2012 and PASCAL Context to quantify the contribution of each core module. As shown in Table 6, adding only the ORA module (row 2; object features randomly initialized) improves unseen mIoU by +1.4 on VOC and +2.0 on Context over the baseline. Enabling DINO-guided prompts (row 3) supplies richer object features to ORA, mitigating poor-initialization effects noted in Slot Attention (Zhang et al., 2023) and yielding further gains on seen and unseen splits. Finally, incorporating granular atten-

tion further improves both seen and unseen performance, achieving the highest scores across both datasets. These results confirm that all components are required for better generalization.

**Comparing Mask Generation Strategies.** We compare strategies for generating attention masks that guide the object prompts (Table 7). SSL-based clustering (DINO-B/16, DINOv2-B/14, DINO-B/8) achieves the best hIoU, with DINO-B/16 slightly outperforming DINOv2, and surpasses masks from CLIP features (Radford et al., 2021) and fixed Region prompts from CLIPRC (Zhang et al., 2024). This trend underscores the need for good feature initialization (Zhang et al., 2023) relative to *No Mask* (random-init object features in ORA). This supports leveraging external self-supervised representations to initialize more effective object prompts.

## 6  LIMITATIONS AND FUTURE WORK.

OC-ZSS relies on an external SSL backbone (DINO in our experiments) to provide object cues, which introduces extra computation and can propagate biases when masks are strongly misaligned with true objects. Although our Voronoi clustering and ORA refinement mitigate some over-/under-coverage issues, heavily overlapping or highly ambiguous objects remain challenging, and gains on near-saturated categories are sometimes modest. In addition, we use fixed global hyperparameters for the number of object prompts and ORA refinement iterations, without adapting them per image or dynamically re-estimating/splitting masks during refinement. Exploring SSL-free ways to guide object prompts or extract object features, as well as improved refinement architectures, is an important direction for future work. Further, adaptive schemes (e.g., entropy-based) for choosing the number of object prompts and iterations, and mechanisms that can re-split or update object masks online, are promising avenues to enhance robustness and flexibility.

## 7  CONCLUSION

We presented OC-ZSS, a novel object-centric framework for zero-shot semantic segmentation that operates entirely on a frozen CLIP backbone. By introducing self-supervision-guided object prompts and a dual-stage Object Refinement Attention (ORA) module, our approach captures and refines object-level features without requiring additional supervision or encoder tuning. Through iterative refinement and multi-scale attention, OC-ZSS promotes stronger semantic grouping in patch representations and improves alignment with textual concepts. Our method achieves state-of-the-art performance across multiple benchmarks and settings, demonstrating that object-centric patch refinement is a powerful yet efficient strategy for advancing zero-shot segmentation.

## 8  REPRODUCIBILITY STATEMENT

All details required to reproduce our results are provided in the Experiments section and Appendix (Implementation Details), including datasets, hyperparameters, training schedules. Upon acceptance, we will release code, configuration files, and checkpoints to enable exact replication.

## ACKNOWLEDGEMENTS

This work was supported by the EuroHPC Joint Undertaking through the EuroHPC Development Access Call (Proposal ID: EHPC-DEV-2025D10-089) and the EuroHPC Benchmark Access Call (Proposal ID: EHPC-BEN-2025B10-041). We are grateful to Umar Marikkar, Iqra Farooq, and Silpa Vadakkeeveetil (University of Surrey), and Carl Udora (University of Bristol), for their valuable feedback during the rebuttal process/compute.

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

## A    MORE IMPLEMENTATION DETAILS

OC-ZSS uses CLIP ViT-B/16 as the backbone, keeping both the text encoder and DINO-B/16 SSL visual encoder frozen. Visual Prompt Tuning (VPT) is applied to the image encoder using 10 learnable prompts. The decoder is a 3-layer lightweight transformer with Multi-Prompt Sinkhorn Attention (MPSA). We set 6 object prompts for VOC and COCO-Stuff, and 8 for PASCAL Context. The Object-Centric Refinement module performs 4 dual-stage refinement iterations by default; for PASCAL Context, we use 5 iterations during both training and testing to handle its higher class diversity. All models are trained using the AdamW optimizer with a learning rate of 2.5e-5, weight decay of 0.01, and batch size of 16. Training is conducted for 20K, 40K, and 80K iterations on VOC 2012, PASCAL Context, and COCO-Stuff, respectively. All experiments are run on a single NVIDIA A100 or four RTX 3090 GPUs. Evaluation is performed using single-scale inference. The values for number of object prompts $no$, number of ORA iterations $S$ is selected based on grid search for each where the values for VOC are shown in further sections.

We compare against standard baselines in both inductive and transductive settings, using reported numbers or official results: ZegFormer (Ding et al., 2022), ZSSeg (Xu et al., 2022b), MaskCLIP+ (Zhou et al., 2022a), ZegCLIP (Zhou et al., 2023), CLIP-RC (Zhang et al., 2024), Cascade-CLIP (Li et al., 2024), OT-Seg (Kim et al., 2024b), MVP-SEG+ (Guo et al., 2023), SPT-SEG (Xu et al., 2024b), MAFT (Jiao et al., 2023), and FreeSeg (Qin et al., 2023). For fairness, we report MAFT combined with ZSSeg in the inductive table, instead of FreeSeg (Qin et al., 2023). FreeSeg adopts a more complex Mask2Former-style query mask design (Cheng et al., 2022) with a broader multi-task recipe for unified/open-vocabulary segmentation, is included only in our transductive comparison, following prior works (Zhang et al., 2024; Li et al., 2024). Before applying the multi-scale adapter, we reshape CLIP patch tokens back to an $H \times W$ grid so that the depthwise atrous convolutions operate on contiguous spatial neighborhoods. The adapter is implemented as parallel $3 \times 3$ depthwise convolutions with fixed dilation rates $\{1, 6, 12, 18\}$, and a $1 \times 1$ projection to produce the final Q/K/V projections. Both the CLIP and DINO encoders are kept *frozen* in all experiments; the only trainable components we add are ORA and visual prompts. Finally, the optimal-transport decoder (including the Sinkhorn iterations, $\varepsilon$ temperature, and log-domain stabilization) is taken *unchanged* from OTSeg, so any performance differences stem from our refinement modules rather than from changes to the decoder. The GRU used in ORA is shared for every refinement iteration including cross-attention layers. All code, hyperparameters, checkpoints will be made available on acceptance.

## B    DECODING PROCESS WITH MPS AND MPSA

OTSeg (Kim et al., 2024b) introduces a decoding strategy that combines Optimal Transport-based prompt-pixel alignment with a transformer-style attention mechanism. It consists of two key components: the **Multi-Prompts Sinkhorn (MPS)** module and the **Multi-Prompts Sinkhorn Attention**

---

**Algorithm 1** Mask generation from DINO last-layer features (FPS Voronoi + promptpatch wiring)

---

**Input:** $F \in \mathbb{R}^{N \times C}$ (DINO last-layer per image), clusters $K$, #VPT tokens $P$, (opt.) target spatial size $S$
**Output:** $M$ (attention mask), labels $\ell$

1: **if** $S$ given **then**
2:    reshape $F$ to $S \times S \times C$, bilinear upsample to $S \times S$, flatten to $N \times C$
3: **end if**
4: **FPS:** pick $c_1$ uniformly; **for** $t=2..K$: $d_i \leftarrow \min_{j<t} \|F_i - F_{c_j}\|_2^2$; $c_t \leftarrow \arg\max_i d_i$
5: **Voronoi labels:** $\ell_i \leftarrow \arg\min_{t \in \{1..K\}} \|F_i - F_{c_t}\|_2^2$ for all $i$
6: **Layout:** total length $A \leftarrow 1+P+N+K$
7: **indices:** $\texttt{cls} = 0$; $\texttt{vpt} = 1{:}P$; $\texttt{patch} = P{+}1{:}P{+}N$; $\texttt{prompt} = P{+}N{+}1{:}P{+}N{+}K$
8: **Init:** $M \leftarrow -\infty \cdot \mathbf{1}_{A \times A}$
9: **for** $i = 1$ **to** $K$ **do**
10:    $\mathcal{P}_i \leftarrow \{ p \in \texttt{patch} \mid \ell_{p-P} = i \}$                           ▷ patch tokens in cluster $i$
11:    Unmask: $M[\texttt{prompt}(i), \mathcal{P}_i] \leftarrow 0$; $M[\mathcal{P}_i, \texttt{prompt}(i)] \leftarrow 0$; $M[\texttt{prompt}(i), \texttt{prompt}(i)] \leftarrow 0$
12: **end for**
13: **return** $M, \ell$

---

(MPSA) module, both aimed at enhancing multimodal alignment between pixel embeddings and prompt embeddings.

**Multi-Prompts Sinkhorn (MPS).**  Given $M$ pixel embeddings $\mathbf{X} \in \mathbb{R}^{M \times d}$ and $N$ text prompt embeddings $\mathbf{T} \in \mathbb{R}^{N \times d}$, MPS first computes a cosine similarity matrix:

$$\mathbf{S} = \mathbf{T} \cdot \mathbf{X}^\top,$$

and defines a cost matrix $\mathbf{C} = 1 - \mathbf{S}$. To align pixels with prompts, OTSeg solves an entropy-regularized optimal transport problem:

$$\mathbf{P}^* = \arg\min_{\mathbf{P} \in \Pi(\mathbf{a},\mathbf{b})} \langle \mathbf{P}, \mathbf{C} \rangle - \varepsilon H(\mathbf{P}),$$

where $H(\mathbf{P})$ is the entropy of the transport plan, and $\Pi(\mathbf{a}, \mathbf{b})$ denotes the set of joint distributions with given marginals. The optimal transport plan $\mathbf{P}^*$ is computed using the Sinkhorn algorithm.

The final prompt-guided score map is obtained as:

$$\widetilde{\mathbf{S}} = \mathbf{P}^* \odot \mathbf{S},$$

which acts as a refined segmentation prediction.

**Multi-Prompts Sinkhorn Attention (MPSA).**  To integrate prompt-pixel transport directly into transformer decoding, OTSeg replaces standard cross-attention with MPSA. Unlike softmax-based attention:

$$\text{Attention}(\mathbf{Q}, \mathbf{K}, \mathbf{V}) = \text{softmax}(\mathbf{Q}\mathbf{K}^\top) \cdot \mathbf{V},$$

MPSA computes the transport matrix via Sinkhorn:

$$\text{MPSA}(\mathbf{Q}, \mathbf{K}, \mathbf{V}) = \text{Sinkhorn}(\mathbf{Q}\mathbf{K}^\top) \cdot \mathbf{V}.$$

This approach promotes diverse prompt activation and better semantic separation in the generated attention maps.

**Decoder Design.**  The decoder used in OTSeg consists of three transformer layers, each applying MPSA instead of standard multi-head cross-attention. The final output is upsampled using a bilinear interpolation operator $\mathcal{U}$ to match the original image resolution:

$$\mathbf{Y}_{\text{final}} = \mathcal{U}(\mathbf{Y}),$$

where $\mathbf{Y} \in \mathbb{R}^{H \times W \times K}$ and $K$ is the number of semantic classes. This decoder output can be used independently or fused with the MPS-based prediction for improved performance.

Table 8: Effect of different design choices for ORA.

| Variant | mIoU(U) | mIoU(S) | hIoU |
|---|---|---|---|
| ORA | **85.3** | **93.6** | **89.2** |
| Slot Attention | 83.4 | 93.1 | 87.9 |
| ORA w/o GRU | 83.1 | 90.9 | 86.8 |
| $1\times$ Cross Attention | 82.6 | 93.4 | 87.7 |
| $2\times$ Cross Attention | 82.2 | 93.2 | 87.3 |
| $4\times$ Cross Attention | 81.6 | 92.9 | 86.8 |

Table 9: Pascal Context with very few seen classes.

| Methods | Context 25% | | | Context 50% | | |
|---|---|---|---|---|---|---|
| | U | S | hIoU | U | S | hIoU |
| CLIP-RC (Zhang et al., 2024) | 11.5 | 51.1 | 18.8 | 18.0 | 45.7 | 25.8 |
| OTSeg (Kim et al., 2024b) | 11.4 | 57.2 | 19.0 | 21.2 | 54.4 | 30.5 |
| OC-ZSS | **12.5** | **58.9** | **20.6** | **23.5** | **55.6** | **33.0** |

Table 10: **Ablation on ORA iterations** ($S$) in the inductive setting.

| Iterations $S$ | 0 | 1 | 2 | 3 | **4 (ours)** | 5 | 6 | 7 |
|---|---|---|---|---|---|---|---|---|
| hIoU (%) | 86.1 | 87.2 | 87.7 | 88.0 | **89.2** | 88.9 | 87.6 | 87.8 |

Table 11: End-to-end throughput (iterations/sec; higher is better).

| Config | it/s ↑ |
|---|---|
| Baseline (OTSeg) | 15.8 |
| Baseline + Object Prompts | 15.2 |
| Baseline + ORA | 13.9 |
| Baseline + Granular Attn | 13.0 |

## C   VORONOI AND MASK GENERATION ON DINO FEATURES.

Given last-layer DINO tokens $F \in \mathbb{R}^{N \times C}$ per image (optionally resized to a fixed grid), we select $K$ centers by farthest-point sampling in feature space and assign each patch token to its nearest center (squared $\ell_2$), inducing a Voronoi partition (Aurenhammer, 1991). We then form the sequence [CLS], $\text{VPT}_{1:P}$, $\text{Patch}_{1:N}$, $\text{ObjPrompt}_{1:K}$ and build a sparse attention mask that unblocks only the connections between $\text{ObjPrompt}_i$ and the patch tokens in cluster $i$ (plus the prompt self-edge); all other entries remain $-\infty$. The implementation is GPU-based in PyTorch with tensor indexing (following (Zhang et al., 2022)), so the per-iteration cost is negligible relative to a CLIP forward; empirically, enabling prompts changes throughput from 15.8 to 15.2 it/s, see Table 11.

## D   MORE ABLATIONS AND VISUALISATIONS

**Effect of Different Design Choices for ORA.**   Table 8 presents an ablation study on the design choices of the proposed Object-Region Attention (ORA) module. The full ORA configuration, which includes both object and patch refinement stages, achieves the best performance across unseen, seen, and harmonic mean IoU. Replacing ORA with Slot Attention (Locatello et al., 2020), which refines patch features from coarse object representations without refining objects first, leads to a noticeable drop, particularly in unseen class performance. Removing the GRU module from ORA (ORA w/o GRU) also reduces performance, validating the role of recurrent refinement. Lastly, we compare against standard cross-attention modules with 1, 2, and 4 stacked layers. Increasing the number of cross-attention layers does not improve performance and instead results in gradual degradation, highlighting the importance of both object and patch refinement stages.

**Ablation on number of ORA iterations.**   We vary the number of refinement iterations (Table 10) $S \in \{0, \ldots, 7\}$ (with $S{=}0$ disabling ORA). hIoU improves up to $S{=}4$ (from 86.1 to 89.2, +3.1), showcasing improved generalisation, then plateaus and slightly declines for larger $S$ (88.9 at 5, 87.6 at 6, 87.8 at 7), suggesting overfitting and reduced generalisability. We therefore fix $S{=}4$ in the main method as an accuracy/latency sweet spot.

**Runtime/throughput ablation.** We report end-to-end throughput (iterations/sec; higher is better) for each configuration under identical hardware and resolution where DINO features are extracted and stored for all images. Relative to the baseline of OTSeg (15.8 it/s), Object Prompts incur a small overhead (15.2 it/s; 3.8% drop; includes clustering), ORA is moderate (13.9 it/s; 12.0% drop), and Granular Attn is the heaviest (13.0 it/s; 17.7% drop). Note that when DINO extraction is performed sequentially, the throughput decreases accordingly due to the additional feature-extraction overhead. This value is provided for reference only and does not represent practical real-world operation, where

Table 12: OC-ZSS with different CLIP vision backbones/sources (inductive setting).

| Backbone Weights / Source | | hIoU (%) |
|---|---|---|
| ViT-B/16 | OpenAI CLIP (Radford et al., 2021) | 89.2 |
| ViT-B/16 | MetaCLIP (Xu et al., 2024a) | 90.1 |
| ViT-L/14 | OpenAI CLIP (Radford et al., 2021) | 90.3 |

Table 13: Attention-mask sources for object prompts (inductive). * is our default setting

| Mask source | hIoU (%) |
|---|---|
| No Mask | 87.1 |
| *DINO-B/16 (Caron et al., 2021) | **89.2** |
| iBoT (Zhou et al., 2022b) | 88.3 |
| DINOv2-B/14 + Reg (Darcet et al., 2024) | **89.2** |

Table 14: Ablation on the number of object prompts $n_o$ on VOC. A moderate number of prompts yields the best trade-off, and we therefore adopt $n_o = 6$ as the default.

| Object prompts $n_o$ | 2 | 3 | 4 | 5 | **6 (ours)** | 7 | 8 |
|---|---|---|---|---|---|---|---|
| hIoU (%) | 87.2 | 87.4 | 88.3 | 88.8 | **89.2** | 88.1 | 87.8 |

DINO feature extraction is executed in parallel across future batches to avoid bottlenecks. As shown in Table 11, these overheads remain modest; taken with the accuracy gains in the main ablation, Table 6, they support including each module in the final model.

**Effect of different CLIP models** We vary the CLIP vision backbone while keeping OC-ZSS fixed DINO-B/16 masks and similar number of object prompts. As shown in Table 12, MetaCLIP-B/16 (Xu et al., 2024a) improves over OpenAI CLIP-B/16 (Our default) (Radford et al., 2021) (89.2 → 90.1 hIoU), and OpenAI CLIP-L/14 yields a further gain of 90.3 hIoU. These results suggest that our object-centric refinement transfers across CLIP variants and backbones. Larger backbones and better variants perform better.

**More experiments on Attention Mask** As in Table 13, *SSL-derived* masks consistently outperform *No Mask* (random initialization), confirming that better feature initialization improves guidance for ORA. In particular, DINO-B/16 and DINOv2 Registers (Darcet et al., 2024) reach 89.2 hIoU, while iBoT attains 88.3, all exceeding the 87.1 hIoU obtained without masks. We use DINO-B/16 instead of DINOv2-B/14 with registers (Darcet et al., 2024) due to better efficiency from smaller patch size of DINO-B/16. This aligns with the trend observed Main paper, Table 7: self-supervised models provide stronger spatial grouping cues for mask generation than unguided initialization.

**Generalization with Limited Seen Classes on Pascal Context.** We evaluate OC-ZSS under extreme low-shot settings on Pascal Context by restricting the number of seen classes to only 25% or 50% during training, compared to the standard zero-shot setting which uses 49 seen and 10 unseen classes out of 59 total. This dataset is inherently more challenging than PASCAL VOC due to its larger number of classes, higher scene complexity, and increased semantic overlap between categories. As shown in Table 9, the performance on unseen classes tends to stagnate for other methods under the 25% setting, with minimal or no gains. Despite this, OC-ZSS still improves over both CLIP-RC and OTSeg, achieving the best unseen and harmonic IoU. These results demonstrate the robustness of OC-ZSS and its ability to generalize to complex and varied unseen categories, even with severely limited supervision.

**Effect of the number of object prompts.** We also vary the number of object prompts $n_o$ on VOC while keeping all other settings fixed. Table 14 reports the harmonic mean (hIoU) between seen and unseen mIoU. Performance improves when moving from very few prompts ($n_o = 2, 3$) to a moderate number ($n_o = 4, 5$), peaks at $n_o = 6$, and then degrades slightly for larger values.

Table 15: Very-few-seen class ablation on seen / unseen IoU. We disentangle the benefit of each major component (object prompts, ORA, and alternative aggregation mechanisms) and how they affect low-shot generalization. All numbers are percentages.

| Method | Unseen IoU | Seen IoU | hIoU |
|---|---|---|---|
| Baseline (OTSeg replication) | 17.70 | 88.30 | 29.40 |
| Baseline + object prompts (SSL guided) + No ORA | 17.40 | 88.75 | 29.10 |
| Baseline + object prompts (SSL guided) + ORA | **49.80** | **92.70** | **64.80** |
| Baseline + ORA (random init object feats) + no object prompts | 42.69 | 89.67 | 57.84 |
| Baseline + object prompts (mask from CLIP) + ORA | 49.40 | 90.39 | 63.89 |
| Baseline + object prompts (SSL guided) + slot attention | 38.35 | 70.85 | 49.76 |
| Baseline + object prompts (SSL guided) + cross-attention | 36.97 | 90.04 | 52.42 |

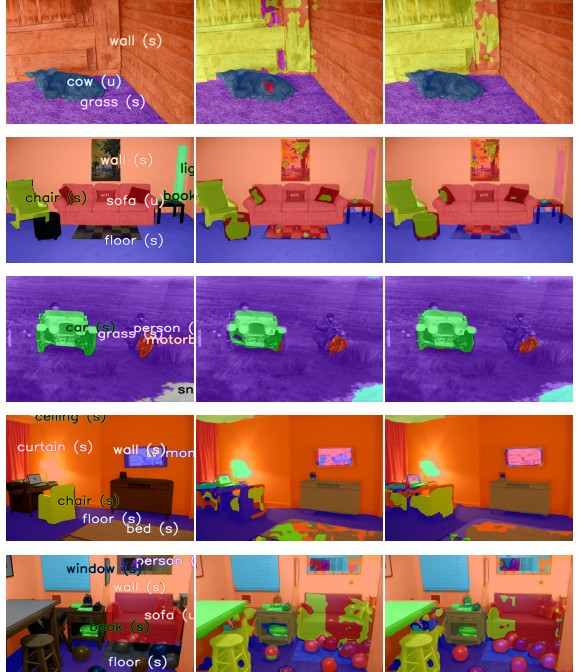

Figure 6: More Mask predictions: input image with ground truth mask (left), OTSeg prediction (middle), and OC-ZSS prediction (right).

Figure 7: More Visualisation: Input image (left), clustered patches without ORA (middle), and with ORA (right).

**Additional Qualitative Examples.** Figure 6 presents additional comparisons between the predictions of OTSeg (Kim et al., 2024b) and our OC-ZSS across diverse scenes. Consistent with the observations in the main paper, OC-ZSS produces more accurate and spatially coherent segmentation masks, particularly for complex object boundaries and less common categories. In contrast, OTSeg predictions often exhibit over-segmentation or coarse localization. Figure 7 further visualizes the impact of the Object-aware Refinement and Aggregation (ORA) module. When ORA is removed, the resulting patch clusters remain noisy and dispersed. With ORA enabled, patches align more clearly with object contours, resulting in more compact and semantically consistent groupings. These examples qualitatively support the effectiveness of our design choices in improving both segmentation quality and region-level consistency.

**Further analysis.** While our main comparisons on VOC, PASCAL Context and COCO-Stuff and the ablations on components, mask sources, and CLIP backbones already show that stronger object features combined with ORA systematically improve zero-shot segmentation, here we study the same components in a more controlled very-few-seen regime to better understand *how* they affect

Table 16: Example per-class IoU on COCO-Stuff comparing OTSeg and OC-ZSS. OC-ZSS brings larger gains on several fine-grained or cluttered categories (top rows), while improvements on near-saturated or ambiguous categories (bottom rows) are smaller.

| Class | OTSeg IoU | OC-ZSS IoU | Δ |
|---|---|---|---|
| wall-concrete | 17.96 | 30.75 | +12.79 |
| skateboard | 40.17 | 46.31 | +6.14 |
| baseball bat | 33.87 | 37.41 | +3.54 |
| stairs | 17.93 | 20.55 | +2.62 |
| road | 38.45 | 42.12 | +3.67 |
| clouds | 32.90 | 34.12 | +1.22 |
| giraffe | 68.17 | 68.85 | +0.68 |
| elephant | 87.84 | 87.95 | +0.11 |

Table 17: Reported performance of recent open-vocabulary segmentation methods on VOC, PAS-CAL Context, and COCO-Stuff. Rows above the line report *mIoU* under their respective open-vocabulary protocols; the OC-ZSS row ([†]) reports *transductive hIoU* under the standard zero-shot segmentation setting with seen/unseen splits. Protocols, training data, and supervision differ substantially from our setting, so these numbers are not directly comparable to our inductive/transductive results.

| Method | VOC | PASCAL Context | COCO-Stuff |
|---|---|---|---|
| SAM-CLIP (Wang et al., 2024) | 60.6 | 29.2 | 31.5 |
| CLIP-DIY (Wysoczańska et al., 2024a) | 59.9 | – | – |
| CLIP-DINOiser (Wysoczańska et al., 2024b) | 62.1 | 32.4 | 24.6 |
| ProxyCLIP (Lan et al., 2024) | 61.3 | 35.3 | 26.5 |
| OVSegmentor (Xu et al., 2023) | 53.8 | 20.4 | – |
| FOSSIL (Barsellotti et al., 2024) | – | 35.8 | 24.8 |
| SimZSS (Stegmüller et al., 2025) | 58.4 | 37.2 | – |
| OC-ZSS (ours, ZSS hIoU)[†] | 95.2 | 61.9 | 51.5 |

generalization. Table 15 decomposes the contributions of (i) SSL/CLIP-guided object prompts, and (ii) the ORA refinement itself.

Starting from our OTSeg baseline (17.70% unseen / 88.30% seen / 29.40 hIoU), adding SSL-guided object prompts *without* ORA leaves unseen IoU and hIoU almost unchanged (17.40% unseen / 88.75% seen / 29.10 hIoU), indicating that stronger object features alone are not sufficient, as the patches are not yet refined with these object features. In contrast, ORA with *randomly initialized* object features already yields a large boost in unseen IoU and hIoU (42.69% unseen / 89.67% seen / 57.84 hIoU), showing that the refinement mechanism itself is crucial for low-shot generalization. Guiding ORA with SSL masks (DINO) further improves performance to 49.80% unseen / 92.70% seen / 64.80 hIoU, and replacing SSL masks with CLIP masks yields similar gains of 49.40% unseen / 90.39% seen / 63.89 hIoU, confirming that our framework is not tied to a particular encoder but consistently benefits from better object features.

Finally, when we keep the same SSL-guided object prompts but replace ORA with slot attention or cross-attention, we obtain clearly weaker trade-offs: slot attention reaches 38.35% unseen / 70.85% seen / 49.76 hIoU, while cross-attention reaches 36.97% unseen / 90.04% seen / 52.42 hIoU. Together with the main-paper results and ablations, these experiments show that (i) We need stronger object features, (ii) making patch representations more object-centric is crucial, and (ii) ORAs design exploits strong object prompts more effectively than standard slot- or cross-attention, leading to robust and balanced generalization in both the standard and very-few-seen regimes.

**Per-class analysis on COCO-Stuff.** To better understand how refinement behaves beyond aggregate mIoU, we report per-class IoU on COCO-Stuff for a few representative fine-grained / cluttered categories and near-saturated or ambiguous classes. As shown in Table 16, OC-ZSS yields substantial gains on several challenging structures where they are thin, small or finegrained (e.g., *wall-concrete*, *skateboard*, *baseball bat*, *stairs*, *road*), while improvements on already high-performing or ambiguous categories (e.g., *clouds*, *giraffe*, *elephant*) are understandably modest.

**Comparison with open-vocabulary segmentation methods**

For completeness, we report a coarse comparison to several recent open-vocabulary (OV) segmentation methods on shared datasets, using the mIoU numbers reported in their respective papers. Note that these results are *not* directly comparable to our transductive ZSS hIoU, since OV methods typically train on large imagetext corpora and do not follow the standard seen/unseen ZSS protocol.

# E   LLM USAGE STATEMENT

We used a large language model only for editorial assistance: polishing prose, correcting grammar and typography, tightening phrasing, and standardizing terminology. The tool was not used to generate ideas, methods, analyses, figures, tables, or results, and it was not used to write code or modify data.

