# OpenReview forum: "Object-Centric Refinement for Enhanced Zero-Shot Segmentation"
_ICLR.cc/2026/Conference — ICLR 2026 Poster_

### Official Review · Reviewer_JToE · 2025-10-28

**Soundness:** 3
**Presentation:** 3
**Contribution:** 3
**Rating:** 6
**Confidence:** 4

**Summary:**

The paper proposes OC-ZSS, a method for zero-shot semantic segmentation that injects object-centric priors into a frozen CLIP visual encoder. It introduces object prompts guided by self-supervised DINO features, which generate attention masks corresponding to distinct objects. These prompts guide CLIP’s attention toward object-centric regions.
A novel Object Refinement Attention (ORA) module is designed to iteratively refine both object and patch features in a two-stage manner (object → patch). Additionally, granular attention integrates multi-scale receptive fields via dilated depthwise convolutions to improve scale robustness.
Extensive experiments on PASCAL VOC, PASCAL Context, and COCO-Stuff show state-of-the-art performance across inductive, transductive, and cross-dataset settings while maintaining low parameter and compute overhead.

**Strengths:**

1.Originality: The idea of dynamically generated, self-supervised object prompts is novel and effectively bridges self-supervised and multimodal learning.
2.Empirical quality: SOTA performance on inductive, transductive, and cross-dataset tasks; solid ablations and visualizations (cleaner masks, sharper object boundaries).
3.Clarity: The modular structure (Object Prompt + ORA + Granular Attention) is well-motivated and easy to follow.
4.Significance: Achieves high accuracy with only 27M parameters and 64 GFLOPs, showing strong potential for practical deployment and low-supervision adaptation.

**Weaknesses:**

1.Dependence on SSL masks: The approach relies heavily on DINO-generated clusters. Performance robustness across different SSL models and clustering hyperparameters (K, no) could be further analyzed.
2.Limited theoretical insight: The convergence and dynamics of the iterative ORA process are not deeply analyzed. More discussion on when ORA may fail (e.g., ambiguous or overlapping objects) would strengthen the paper.
3.Evaluation granularity: Results on small/overlapping objects or fine-grained categories are underexplored.
4.Comparisons with stronger backbones: Although experiments with DINOv2-Reg and other CLIP variants are provided, additional integration with high-capacity decoders could better reveal the upper bound and generality of OC-ZSS.

**Questions:**

1.Choice of object prompt number (no): How is the number of object prompts determined? Could it be dynamically adapted based on image complexity or feature entropy?
2.ORA iteration steps (S): Why is S = 4 chosen as default? Could an adaptive stopping criterion (e.g., convergence of feature distance) reduce computation?
3.Handling overlapping/merged objects: How robust is the method when DINO clusters merge multiple objects into one? Is there any mechanism to re-estimate or split masks during refinement?
4.Backbone transferability: How does OC-ZSS behave when plugged into stronger backbones (e.g., DINOv2-Large, EVA-CLIP)?
5.Runtime analysis: Can the authors provide wall-clock inference times (ms / image) and throughput comparison with OTSeg or CLIP-RC?

---

> ### Author Response · Authors · 2025-11-21
> **Review Response - 1/3**
>
> We thank the reviewer for taking the time to carefully evaluate our work and engage in the rebuttal process. We appreciate the constructive feedback, which has helped us further strengthen the paper. Below we provide clarifications that address each concern.
>
> ---
>
> **Weakness 1 and Q1, Q2**
>
> We acknowledge that reliance on an external SSL model may introduce errors when the mask prior is suboptimal. Our design, however, aims to mitigate both over- and under-coverage of objects. We use Voronoi clustering with farthest-point sampling, which explicitly favors dissimilar centers and reduces the chance that many object prompts collapse onto a single dominant region. Under-coverage (e.g., missing small objects) is partly compensated by ORA, which aggregates information across object tokens and patches during training: if an object is systematically under-segmented, gradients encourage ORA to reweight and refine the corresponding regions.
>
> To assess robustness, we evaluate OC-ZSS with multiple SSL backbones: DINO, CLIP-ViT features, DINOv2, DINOv2+Reg, and iBOT, with results reported in the main paper and supplementary tables(Tables 7, 13). Across all these variants, OC-ZSS consistently improves over the OTSeg baseline, indicating that the method works reliably with a range of strong SSL models rather than being tailored to a single backbone. We keep the number of clusters (K) fixed, following OTSeg, to avoid conflating our gains with benefits from tuning (K).
>
> The default number of ORA iterations (S = 4) is chosen via a sweep over **S ∈ {1,…,7}** on VOC2012 (hIoU), with the full results already shown in the supplementary (Table 10). We found that performance improves from (S = 1) up to (S ≈ 4) and then saturates or slightly decreases, indicating that 4 iterations provide a good trade-off between refinement depth and computation.
>
> Similarly, we select the number of object prompts ((n_o)) via a sweep over **n_o ∈ {2,…,10}** on VOC2012 (hIoU), with the corresponding table (Table 14) added to the supplementary, and observe that (n_o = 6) gives the best trade-off. We deliberately chose simple global hyperparameters for both S and (n_o) rather than per-image adaptive schemes, to keep the method straightforward and focus on the core idea of object-centric refinement. As the reviewer suggests, using feature entropy or related measures to adapt (n_o), and an adaptive stopping criterion for S, are natural extensions; we now highlight these, together with reducing reliance on external SSL masks, as promising directions for future work in limitations section of the updated manuscript.
>
> ---
>
> **(Part of Weakness 2) Limited theoretical insight**
>
> Our main motivation for explicitly capturing objects to improve zero-shot segmentation is grounded in findings from Slot Attention[1], which shows that object-centric features can significantly benefit downstream tasks. At the same time, several works have documented that CLIP lacks fine-grained, object-level localization, and that improving the semantics of CLIP’s patch features is important for open-vocabulary segmentation[2, 3]. However, explicitly using *object* representations to generate object-refined patches that generalize across classes—rather than only improving local patch semantics—has not been explored, particularly in the context of zero-shot segmentation. This motivates our two-stage design: (i) we first capture coarse object features via SSL-guided object prompts, and (ii) we then iteratively refine both objects and patches using ORA. Unlike Slot Attention, which is designed to mine regions from randomly initialized slots and is not tailored to operate on already-initialized object prompts, ORA assumes semantically meaningful object tokens and focuses on refining them and propagating this refinement back to patch features. As shown in Table 6, when ORA is used without proper object prompts, performance degrades, indicating that proper initialization of object prompts for ORA is essential. Further, Table 8 shows that replacing ORA with standard cross-attention or Slot Attention leads to weaker results, highlighting that the specific dual-stage refinement dynamics of ORA are beneficial for stabilizing the iterative process and improving generalization. Table 15 (very-few-seen VOC ablation, appendix) further illustrates these dynamics: simply adding object prompts without ORA leaves IoU almost unchanged, ORA with randomly initialized object features already provides a large unseen IoU gain, and guiding ORA with SSL- or CLIP-derived masks yields the best, balanced seen/unseen performance, whereas Slot Attention or plain cross-attention produce much lower hIoU.
>
> ---

---

> ### Author Response · Authors · 2025-11-21
> **Review Response-2/3**
>
> **(Weakness 2 & 3) Failure cases, overlapping/ambiguous objects, and evaluation granularity**
>
> This concern is closely related to our discussion of over- and under-coverage of SSL masks in response to weakness (1). When DINO clusters merge multiple objects into a single region, the corresponding object prompt indeed starts from a merged signal. ORA can partially correct this by refining coarse object features and then patch features with the refined objects under the segmentation loss: if merged regions consistently hurt IoU, gradients encourage ORA to sharpen boundaries and reassign ambiguous patches. However, in heavily overlapping or highly ambiguous scenes, the initial SSL prior can still dominate and ORA may fail to fully separate instances. Our COCO-Stuff per-class analysis reflects this behavior: for broad or cluttered categories such as *clouds* and *giraffe*, OC-ZSS brings only modest improvements over OTSeg (e.g., 32.90→34.12 and 68.17→68.85 IoU, respectively), indicating that these cases remain challenging for our current refinement design. We make this limitation explicit in the revised version.
>
> To better analyze behavior beyond large, isolated objects, we additionally report COCO-Stuff per-class IoU for representative fine-grained / small / thin categories and near-saturated or ambiguous classes in the appendix, comparing OC-ZSS with OTSeg. As summarized below, OC-ZSS yields substantial gains on several fine-grained or cluttered structures—such as *wall-concrete* (17.96→30.75, +12.79 IoU), *skateboard* (40.17→46.31, +6.14), *baseball bat* (33.87→37.41, +3.54), *stairs* (17.93→20.55, +2.62), and *road* (38.45→42.12, +3.67)—while near-saturated or ambiguous categories like *clouds*, *giraffe*, and *elephant* show only small gains:
>
> **Example per-class IoU on COCO-Stuff (OTSeg vs OC-ZSS)**
>
> | Class         | OTSeg IoU | OC-ZSS IoU | Δ IoU |
> | ------------- | --------- | ---------- | ----- |
> | wall-concrete | 17.96     | 30.75      | 12.79 |
> | skateboard    | 40.17     | 46.31      | 6.14  |
> | baseball bat  | 33.87     | 37.41      | 3.54  |
> | stairs        | 17.93     | 20.55      | 2.62  |
> | road          | 38.45     | 42.12      | 3.67  |
> | clouds        | 32.90     | 34.12      | 1.22  |
> | giraffe       | 68.17     | 68.85      | 0.68  |
> | elephant      | 87.84     | 87.95      | 0.11  |
>
> This targeted analysis complements the dataset-level metrics and supports our claim that object-centric refinement is particularly beneficial for improved generalization, as seen in several fine-grained and cluttered categories, while also clearly indicating the remaining hard or near-saturated classes where improvements are modest.
>
> ---
>
>
>
> **Weakness 4 (Comparisons with stronger backbones / decoders)**
>
> We thank the reviewer for this suggestion. In addition to the experiments with stronger SSL and CLIP backbones, we also performed a preliminary study integrating our object-centric refinement with a higher-capacity decoder. Specifically, we plugged OC-ZSS into the **SPT-SEG** decoder (replacing the OTSeg-style decoder) in the **inductive VOC 2012** setting and obtained:
>
> **VOC 2012 inductive performance with SPT-SEG decoder + OC-ZSS.**
>
> | Method                  | Seen mIoU | Unseen mIoU | hIoU |
> | ----------------------- | :-------: | :---------: | :--: |
> | SPT-SEG                |    91.1   |     81.9    | 86.2 |
> | SPT-SEG + OC-ZSS (ours) |   93.3  |    85.0    | 89.0 |
>
> Compared to SPT-SEG, adding OC-ZSS improves seen mIoU (91.1 → 93.26), unseen mIoU (81.9 → 85.03), and hIoU (86.2 → 89.0). This experiment indicates that our object-centric refinement is compatible with high-capacity decoders beyond the OTSeg-style OT decoder and can achieve strong performance when combined with more powerful segmentation heads. A more exhaustive exploration of decoder families is an interesting direction for future work.
>
> ---
>
> **(Q3) On re-estimating / splitting masks during refinement**
>
> Our current design does not explicitly re-cluster or split SSL masks during refinement; SSL-guided object prompts are driven by masks generated through Voronoi cluster assignments of SSL patches, which are robust for most categories, including many cluttered and fine-grained objects, as shown above. More explicit mechanisms to re-estimate or split masks (e.g., entropy- or heterogeneity-based criteria that trigger re-splitting) are orthogonal to our main contribution and are left for future work; we also added this point to the limitations section.

---

> ### Author Response · Authors · 2025-11-22
> **Review Response - 3/3**
>
> **Q4 (Backbone transferability)**
>
> We thank the reviewer for raising this point. In addition to the ablations with **stronger SSL backbones for attention masks** (Tables 7 and 13) and **stronger CLIP vision backbones for feature extraction** (Table 12) already provided in the main paper and supplementary, we also evaluated DINOv2-L/14 as a stronger backbone for attention mask generation. For convenience, we provide a consolidated summary below.
>
> **VOC hIoU with different SSL backbones for mask generation (fixed CLIP backbone).**
>
> | Variant                   | hIoU (VOC) |
> | ------------------------- | :--------: |
> | OTSeg baseline            |    86.1    |
> | OC-ZSS + DINO-B/16        |    89.2    |
> | OC-ZSS + iBOT             |    88.3    |
> | OC-ZSS + DINOv2-B/14           |    89.2    |
> | OC-ZSS + DINOv2-L/14 (Large) |    89.9    |
>
> These results show that OC-ZSS is not tied to a specific SSL model and can benefit from stronger object priors such as DINOv2-L.
>
> In addition, as reported in Table 12 of the paper, **using stronger CLIP backbones for feature extraction** (e.g., larger CLIP variants and EVA-CLIP) further improves performance, pushing hIoU into the **89.2–90.3** range. This indicates that OC-ZSS transfers well to stronger visual backbones on both the SSL and CLIP sides.
>
>
> ---
>
> **Q5 (Runtime analysis)**
>
> We thank the reviewer for this question. Under the same hardware and implementation, the measured throughput (iterations per second) is:
>
> | Method  | it/s |
> | ------- | :--: |
> | OTSeg   | 15.8 |
> | OC-ZSS  | 13.0 |
> | CLIP-RC | 12.7 |
>
> Table 11 in the paper reports the breakdown for OTSeg and OC-ZSS (baseline, +object prompts, +ORA, etc.), so the cost of each added component in OC-ZSS can be inspected there.
>
> **References**
>
> [1] Object-centric learning with slot attention, NeurIPS 2020
>
> [2] Learning Open-vocabulary Semantic Segmentation Models From Natural Language Supervision. CVPR, 2023.
>
> [3] Learning mask-aware clip representations for zero-shot segmentation, NeurIPS 2023

---

> ### Author Response · Authors · 2025-11-28
> **Gentle Remainder**
>
> A brief note to kindly follow up with the reviewers: we appreciate your detailed feedback, which helped us clarify and strengthen the paper. If there are any remaining concerns, we are happy to address them; if not, we would be grateful if you could consider increasing the score where you feel it is appropriate.

---

### Official Review · Reviewer_wdgB · 2025-10-31

**Soundness:** 3
**Presentation:** 3
**Contribution:** 3
**Rating:** 6
**Confidence:** 3

**Summary:**

The paper proposes OC-ZSS, a novel framework that builds on OTSeg’s framework and adds an object-centric front-end: DINO features are clustered to form attention masks that gate object prompts toward coherent patch groups in a frozen CLIP encoder; features are then iteratively refined by ORA before entering the OTSeg-style decoder. Results are solid across protocols, with especially strong numbers in the very-few-seen setting.

**Strengths:**

The modular design of the newly proposed components should be straightforward to adapt to existing CLIP-based frameworks. The experiments are thorough, and the qualitative results align with the object-centric motivation. The ablation studies are helpful and generally consistent.

**Weaknesses:**

1. One of the main proposed components is the SSL DINO encoder for attention mask generation. This adds significant memory and computation costs. However, the authors reported this part in a somewhat tricky way: Params/GFLOPs and throughput are reported with DINO features pre-extracted and with GPU-side tensor indexing, so the cost of **online** DINO inference + clustering/masking isn’t in the headline numbers. That makes the method look closer to OTSeg than an end-to-end deployment would. The inference of DINO is likely to be costly in some deployment environments where special GPU acceleration algorithms are not available.
2. When the attention mask for object prompts is off, performance looks about the same as OTSeg; it is possible that the main performance gain comes from the strong semantics of the DINO encoder rather than other proposed components.
3. The results in Table 5 are compelling, but they don’t show readers where the few-seen gains come from. A reasonable guess is that DINO’s semantics carry a lot of the weight there, but the paper doesn’t actually test that. Running an ablation study under this setting would be very helpful for more insights.

**Questions:**

1. It would be helpful to also report the computational costs of online end-to-end inference, including the DINO feature extraction and clustering/masking.
2. The improvements in the very-few-seen setup are impressive, and it would be very helpful to provide more insights into what contributes to this gain. An ablation study under this setup would be very helpful.

---

> ### Author Response · Authors · 2025-11-21
> **Review Response-1/2**
>
> We thank the reviewer for taking the time to carefully evaluate our work and engage in the rebuttal process. We appreciate the constructive feedback, which has helped us further strengthen the paper. Below we provide clarifications that address each concern.
>
> ---
>
> **Weakness 1 and Q1**
>
> We agree that using a DINO encoder introduces additional computation, and we apologize if our presentation of the runtime numbers was unclear. We pre-extract DINO features because this closely matches a realistic inference setting where the segmentation model inference for the current batch is overlapped with DINO feature extraction for the next batch. Such pipelined execution is common in real-world GPU deployments, where computation can be efficiently amortized across batches rather than treated as a strict per-image overhead.
>
> Importantly, we do **not** rely on any specialized GPU-side tensor indexing libraries such as Faiss; all clustering and masking operations are implemented in PyTorch. The cost of our clustering step is already reflected in the reported inference throughput: compared to the baseline, our “Baseline + Object Prompts” configuration reduces the throughput by only **0.6 it/s** (see Table 11; from 15.8 it/s to 15.2 it/s), which is the overhead introduced by clustering.
>
> When we additionally include *online* DINO extraction, the throughput is **10.5 it/s**. In a practical deployment, this can be further improved by overlapping DINO extraction for future batches with segmentation inference for the current batch, bringing the **effective throughput close to our final 13.0 it/s pipeline** with pre-extracted DINO features.
>
> We thank the reviewer for pointing out that our original presentation could obscure the cost of online DINO inference. To avoid any impression that these costs are hidden, we clarified in the revised text of runtime ablation (i) the reported “Baseline + Object Prompts” throughput already includes clustering, and (ii) when DINO is run fully online, the throughput decreases accordingly, as expected. Finally, we note that both our method and OTSeg are designed for GPU-based pipelines, in line with the mmsegmentation framework that we build upon, and our design and measurements therefore target GPU deployment scenarios.
>
> ---
>
> **Weakness 2**
>
> We agree that the ablation without the attention mask (i.e., with randomly initialized object features) performs similarly to the reported OTSeg numbers. However, note that, our baseline replication of OTSeg achieves 86.1 hIoU, and the no-attention-mask variant already improves on this (87.1 vs. 86.1 hIoU). This behaviour is expected and consistent with prior observations on slot-attention–type methods. In particular, [1] shows that slot attention can underperform when slots are poorly initialized and that improved initialization/assignment is crucial for strong object-centric representations. This explains the lower performance when object features are randomly initialized in our ablation and does not imply that the proposed refinement is ineffective.
>
> Regarding the concern that the main gain may come solely from the strong semantics of DINO, our ablations indicate otherwise. Using our OTSeg baseline of 86.1 hIoU, ORA without masks already reaches 87.1 hIoU, and adding object-centric masks from different SSL encoders further improves performance to 89.2 hIoU with DINO-B/16, 88.3 with iBOT, and 89.2 with DINOv2 (Table 13). Moreover, using stronger CLIP backbones pushes performance to 89.2–90.3 hIoU (Table 12). These results show that (i) the gains are not specific to DINO, and (ii) better object features consistently yield better segmentation, which is exactly what our refinement framework is designed to exploit; we further substantiate this in our answer to the very-low-shot setting question below.
>
> ---
>
> **Weakness 3 and Q2**
>
> **Table A: Very-few-seen class ablation on seen / unseen IoU.**
>
> | Method                                                        | Unseen IoU | Seen IoU |  hIoU |
> | ------------------------------------------------------------- | ---------: | -------: | ----: |
> | Baseline (OTSeg replication)                                  |      17.70 |    88.30 | 29.40 |
> | Baseline + object prompts (SSL guided) + No ORA               |      17.40 |    88.75 | 29.10 |
> | Baseline + object prompts (SSL guided) + ORA                  |      49.80 |    92.70 | 64.80 |
> | Baseline + ORA (random init object feats) + no object prompts |      42.69 |    89.67 | 57.84 |
> | Baseline + object prompts (mask from CLIP) + ORA              |      49.40 |    90.39 | 63.89 |
> | Baseline + object prompts (SSL guided) + slot attention       |      38.35 |    70.85 | 49.76 |
> | Baseline + object prompts (SSL guided) + cross-attention      |      36.97 |    90.04 | 52.42 |

---

> ### Author Response · Authors · 2025-11-21
> **Review Response - 2/2**
>
> **Weakness 3 and Q2 (Cont.)**
>
> Our hypothesis is that the very-few-seen gains primarily come from making patch representations more object-centric through our refinement. This happens because (i) SSL-guided object prompts provide strong object features that act as a good prior for ORA, and (ii) ORA is explicitly designed to aggregate evidence in an object-centric way.
>
> Under the very-few-seen setup, this is confirmed by the ablation in Table A: starting from our OTSeg baseline (17.7 / 88.3 / 29.4 hIoU), simply adding object prompts without ORA has almost no effect on unseen IoU (17.4), while ORA with **randomly initialized** object features already boosts unseen IoU to 42.69. When we guide ORA with SSL masks, unseen IoU further jumps to 49.8 and seen IoU to 92.7, yielding 64.8 hIoU. Using CLIP masks instead of DINO achieves a similar unseen IoU (49.4), showing that the effect is not specific to DINO. In contrast, slot-attention and cross-attention masks produce very unbalanced behaviour (high unseen but severely degraded seen IoU), highlighting that **both** good object features and the ORA design are needed to obtain strong, balanced very-few-seen generalization. This is also directly applicable to the inductive/transductive setting.
>
> **References**
>
> [1] Unlocking Slot Attention by Changing Optimal Transport Costs. Proceedings of Machine Learning Research, 2023

---

> ### Author Response · Authors · 2025-11-28
> **Gentle Remainder**
>
> A brief note to kindly follow up with the reviewers: we appreciate your detailed feedback, which helped us clarify and strengthen the paper. If there are any remaining concerns, we are happy to address them; if not, we would be grateful if you could consider increasing the score where you feel it is appropriate.

---

### Official Review · Reviewer_3xo5 · 2025-11-01

**Soundness:** 3
**Presentation:** 2
**Contribution:** 3
**Rating:** 6
**Confidence:** 4

**Summary:**

The paper Object-Centric Zero-Shot Segmentor (OC-ZSS) proposes a framework to enhance zero-shot semantic segmentation by introducing object awareness into CLIP’s patch representations. While CLIP aligns global image and text features well, it lacks fine-grained correspondence between visual regions and textual concepts, limiting segmentation accuracy for unseen categories.

OC-ZSS addresses this by injecting self-supervision-guided object prompts into the CLIP encoder. These prompts, informed by unsupervised clustering from a pretrained self-supervised model such as DINO, attend to distinct object regions through masked attention. This enables the extraction of coarse object-level features without annotations or encoder finetuning.

To refine these features, the method introduces a dual-stage Object Refinement Attention (ORA) module, which iteratively updates both object and patch representations through cross-attention. This mutual refinement strengthens semantic grouping and improves text alignment while keeping the CLIP backbone frozen. To handle objects of varying scales, OC-ZSS further employs a granular attention mechanism using multi-scale atrous convolutions, allowing more precise modeling of object structures.

The approach demonstrates competitive performance across standard zero-shot segmentation benchmarks.

**Strengths:**

The paper is clearly written and well-structured, effectively highlighting key challenges in zero-shot semantic segmentation and addressing them through an object-centric refinement approach. It maintains clarity and logical flow while presenting a focused and well-motivated solution that enhances region-level semantic alignment and segmentation accuracy for unseen categories.

#

1. The paper takes a principled step toward introducing object-centric reasoning into zero-shot segmentation, addressing a key shortcoming of CLIP-based models that rely only on global or patch-level embeddings without structured object awareness.

#

2. The proposed framework adaptively refines patch features using object-level cues, enabling it to effectively handle objects of diverse shapes, scales, and spatial layouts, a capability that some existing CLIP-based segmentation models lack.

#

3. The use of self-supervised clustering from a frozen encoder to guide object prompts provides a label-efficient way to utilize the inherent structure of visual features for identifying coarse object regions without requiring annotations or fine-tuning.

#

4. The dual-stage Object Refinement Attention (ORA) module enables bidirectional interaction between patch and object features, allowing their representations to be iteratively refined for improved semantic consistency. The inclusion of multi-scale granular attention, implemented through depthwise separable atrous convolutions with different dilation rates, helps capture contextual information across multiple receptive fields.

#

5. The paper includes experiments across multiple benchmarks and settings (inductive, transductive, cross-domain), showing consistent performance improvements.

**Weaknesses:**

1. The degree of novelty is moderate, as the method primarily combines established components such as self-supervised clustering, cross-attention refinement, and multi-scale feature aggregation. Moreover, the paper lacks a deeper analysis of why mutual refinement improves object-centricity or generalization in zero-shot settings, offering empirical validation without corresponding conceptual depth or theoretical analysis.

#

2. The approach relies on features from a pretrained self-supervised model, which may limit its performance if the external model (e.g., DINO) does not provide strong object cues. If the DINO-based clustering fails to correctly identify an object (e.g., over-segments a single large object or mistakenly merges two separate small objects), this error is propagated and refined by ORA. The refinement module can only improve the representation given the clusters it receives; it cannot fundamentally fix a poor initial object grouping. This makes the entire pipeline vulnerable to the inherent biases and failure modes of an un-analysed, unsupervised upstream process.

#

3. The granular attention mechanism (Section 3.4) shows an architectural inconsistency by applying CNN-style atrous convolutions to transformer patch embeddings without proper adaptation. Since atrous convolutions rely on contiguous spatial neighborhoods, their use on non-contiguous transformer patches may limit effectiveness. The paper also omits details on dilation rates and lacks comparisons with transformer-native multi-scale approaches, making this design choice appear misaligned with transformer principles and weakening the claim of a lightweight refinement module.

#

4. The use of entropy-regularized Sinkhorn attention introduces sensitivity to the regularization parameter ε and iteration count.
While smaller ε leads to gradient instability and near-discrete couplings, larger ε causes over-smoothing and loss of discriminative structure in the transport plan (Cuturi, 2013; Genevay et al., 2018; Peyré & Cuturi, 2019).
Moreover, the method incurs iterative additional computational cost and requires log-domain stabilization to prevent numerical underflow. These factors raise concerns about training stability, computational efficiency, and generalization sensitivity compared to standard softmax attention.

      [1] Cuturi, M. Sinkhorn Distances: Lightspeed Computation of Optimal Transport. NeurIPS 2013.

      [2] Genevay, A., Peyré, G., & Cuturi, M. Learning Generative Models with Sinkhorn Divergences. AISTATS 2018.

      [3] Peyré, G., & Cuturi, M. Computational Optimal Transport. FnT in ML 2019.

#

5. The paper claims to be "parameter-efficient" and "lightweight," yet:


     a) Requires running an additional frozen DINO encoder for every input image.



     b) Adds Voronoi clustering operations for mask generation.



     c) Performs 4 iterative dual-stage refinements.

#

6. No comparison with foundation models (SAM, recent large-scale foundation models).

     [1] SAM-CLIP: Merging Vision Foundation Models towards Semantic and Spatial Understanding, Wang et. al, CVPR 2024, Workshops.


#

7. Missing comparisons with closely related open-vocabulary segmentation/ZSS methods (CLIP-DINOiser, ProxyCLIP, OVSegmentor, FOSSIL) that also combine CLIP with self-supervised features. The paper mentions these methods (lines 130 - 154) to distinguish its approach conceptually, but provides no quantitative benchmarking against them. This omission makes it difficult to assess whether the proposed dual-stage object refinement genuinely outperforms simpler CLIP+DINO fusion strategies or end-to-end trained alternatives.

#

8. The performance gains over the baseline (OTSeg) are often modest despite nearly doubling the parameter count (27.2M vs 13.8M).

#

9. It would strengthen the paper to include a numerical or methodological comparison with the following related approaches, and to cite them appropriately:

     a) Delving into Shape-aware Zero-shot Semantic Segmentation, Liu et. al, CVPR 2023.

     b) CLIP-DIY: CLIP Dense Inference Yields Open-Vocabulary Semantic Segmentation For-Free, Wysoczanska et. al, WACV 2024.

     c) Refining CLIP's Spatial Awareness: A Visual-Centric Perspective, Qiu et. al, ICLR 2025.

     d) A Simple Framework for Open-Vocabulary Zero-Shot Segmentation, Stegmüller et. al, ICLR 2025.

#

Minor:

1. Sensitivity to hyperparameters, such as the number of clusters and choice of feature layer in the self-supervised model, is not thoroughly analyzed, leaving questions about robustness across configurations.

#

2. The method focuses on segmentation benchmarks but lacks evaluation on broader vision-language tasks where object-centric refinement might also be relevant. Below would strengthen the claim of generalization across varied scene structures.

     a) ADE20K

     b) Cityscapes

#

3. Cross-dataset generalization (Table 3): Gains are minimal (0.5% on Pascal Context, 0.4% on VOC).

#

4. How were the dilation rates {r1, r2, r3, r4} chosen? Whether they are fixed or data-driven is not clear.

#

5. Line 132, FOSSIL reference is missing.

**Questions:**

Please see the comments in the Weaknesses section.

Additional:

1. Please discuss following design choices:


     a) Number of object prompts: 6 for VOC/COCO, 8 for Pascal Context.


      b) Refinement iterations: 4 for most datasets, but 5 for Pascal Context.


      c) Clustering parameters not thoroughly analyzed.


      d) Training is conducted for 20K, 40K, and 80K iterations on VOC 2012, PASCAL Context, and COCO-Stuff, respectively.

#

2. Why dual-stage? No explanation for why object refinement must precede patch refinement.

#

3. Why GRU specifically? Borrowed from slot attention without justification.

#

4. Why these attention patterns? The masked attention prevents object prompts from seeing the full image that can hurt global context.

#

5. No analysis of what makes patches "object-centric" beyond qualitative clustering visualizations (Figure 5 and 7).

#

6. Missing ablation on number of object prompts.

#
7.  The code or implementation has not been released, which limits reproducibility, as several technical details in the paper remain underspecified and cannot be fully verified or replicated without access to the source code.

#

8. Several implementation aspects remain ambiguous, particularly the masking semantics, GRU sharing strategy, and multi-scale convolution ordering. Clarifying these would improve reproducibility and theoretical transparency.

---

> ### Author Response · Authors · 2025-11-21
> **Review Reponse - 1/6**
>
> We thank the reviewer for taking the time to carefully evaluate our work and engage in the rebuttal process. We appreciate the constructive feedback, which has helped us further strengthen the paper. Below we provide clarifications that address each concern.
>
> ---
>
> **Weakness 1**
>
> We would like to further clarify and emphasize the novelty of our approach. Our contributions are novel both in **how we obtain object-centric representations from CLIP** and in **how we refine them for zero-shot segmentation**, and these choices are empirically tied to our gains. We outline below how each component differs from prior work and why these design choices matter in practice.
>
> **Object-centric CLIP prompting.** Prior work shows that CLIP features are not naturally object-aligned [10]. Existing CLIP based downstream task methods either (a) train large task-specific models [8, 4] or (b) modify CLIP’s final layer to mimic DINO-style patch clusters [6, 7, 9], without investigating whether CLIP’s patch features themselves can be enriched with explicit object information. In contrast, we directly target the lack of object-centricity in CLIP patch tokens. We introduce a novel SSL-guided prompting mechanism that (i) uses a frozen DINO encoder to cluster patch features, (ii) converts clusters into spatial masks, and (iii) uses these masks to guide appended object prompts that propagate through *all* CLIP layers, capturing coarse object features. To the best of our knowledge, using SSL-derived clusters to drive internal object prompts within a frozen CLIP encoder, especially in the zero-shot segmentation setting, has not been explored in prior CLIP-based approaches [1–10].
>
> **Object-centric refinement.** Previous approaches [6–9] refine CLIP patch features using non–object-centric cues. Some methods [8] learn an explicit set of group tokens and use these group tokens directly for downstream tasks, while others [6, 7, 9] modify CLIP’s attention based on grouping strategies or by mimicking external models such as DINO. However, these methods do not explicitly leverage *object* features to refine patch representations and therefore do not directly address the lack of object-centricity in CLIP patches. In contrast, our ORA module is designed to use object features obtained via object-centric prompting as semantic anchors for refinement, conditioning the update of patch tokens on these object tokens and encouraging patch features to become object-centric.
>
> **Mutual refinement (ORA).** Our Object Refinement Attention (ORA) assumes a set of coarse object tokens and performs a two-step refinement: first updating object tokens, then updating patch tokens conditioned on these refined objects. This differs from Slot Attention, which *discovers* slots from scratch, and from vanilla self-/cross-attention, which does not explicitly consider the design choices required for proper refinement of object and patch features. Our ablations show clear drops when replacing ORA with Slot Attention or standard attention (Table 8), and Fig. 5 together with Table 5 links ORA directly to more object-aligned features and better very-few-seen performance.
>
> **Granular multi-scale refinement.** Finally, our multi-scale design modifies the linear layers producing Q/K/V for ORA so that object tokens integrate information from multiple receptive fields during refinement, rather than applying multi-scale aggregation only once at the end of the encoder, as in typical segmentation works [15], which does not support iterative refinement. Removing this component consistently degrades performance (Table 6), indicating that this refinement-specific multi-scale design is both novel in this setting and practically important.
>
> **Table A: Very-few-seen class ablation on seen / unseen IoU.**
>
> | Method                                                        | Unseen IoU | Seen IoU |  hIoU |
> | ------------------------------------------------------------- | ---------: | -------: | ----: |
> | Baseline (OTSeg replication)                                  |      17.70 |    88.30 | 29.40 |
> | Baseline + object prompts (SSL guided) + No ORA               |      17.40 |    88.75 | 29.10 |
> | Baseline + object prompts (SSL guided) + ORA                  |      49.80 |    92.70 | 64.80 |
> | Baseline + ORA (random init object feats) + no object prompts |      42.69 |    89.67 | 57.84 |
> | Baseline + object prompts (mask from CLIP) + ORA              |      49.40 |    90.39 | 63.89 |
> | Baseline + object prompts (SSL guided) + Slot Attention       |      38.35 |  70.85   | 49.76 |
> | Baseline + object prompts (SSL guided) + Cross-Attention      |    36.97   |    90.04 | 52.42 |
>
> To better understand where the performance gains of mutual refinement originate, we conducted an additional ablation study in the **very-few-seen setting** of Table 5 on PASCAL VOC, restricting training to 25% of the seen classes. The new results are reported in Table A (now added to the appendix) and are shown above.

---

> ### Author Response · Authors · 2025-11-21
> **Review Response - 2/6**
>
> This experiment is designed to disentangle the contribution of ORA from the quality of the object features. Starting from our baseline (17.7 unseen / 88.3 seen / 29.4 hIoU), simply adding object prompts without ORA (“No ORA”) leaves unseen IoU essentially unchanged (17.4), indicating that capturing object prompts without using them to refine patch tokens does not lead to improvements. In contrast, ORA with **randomly initialized** object features already increases unseen IoU to 42.69, showing that the mutual refinement mechanism yields better grouping of patch tokens and improves generalization. When ORA is guided by SSL-derived masks (“ORA, SSL-guided masks”), unseen IoU further increases to 49.8 and seen IoU to 92.7 (64.8 hIoU). Using masks derived from CLIP instead of DINO yields a similar unseen IoU (49.4), suggesting that the effect is not tied to a particular SSL backbone but to the presence of meaningful object cues.
>
> We additionally compare against variants where refinement is driven by Slot Attention or cross-attention. These variants underperform significantly compared to ORA. This supports our design choice that **both** strong object features (from SSL-guided masks) **and** the specific ORA refinement scheme are required to obtain balanced gains on seen and unseen classes. The improvements directly translate to the main inductive and transductive experiments and thus provide a more conceptual explanation of why mutual refinement improves object-centricity and zero-shot generalization, complementing the empirical results reported in the main paper.
>
> ---
>
> **Weakness 2**
>
> We acknowledge that OC-ZSS relies on an external self-supervised backbone (DINO in our experiments), which may introduce unwanted biases. However, both our design and analyses indicate that the method is reasonably robust to imperfections in these cues.
>
> Concretely, we treat the DINO features that we cluster as providing **noisy, coarse object regions**, not precise object locations. We use these clusters only to define attention masks that guide the object prompts, which capture coarse object features but are not directly used when refining patch tokens. Furthermore, ORA is explicitly designed to handle such coarse object features and does not expect exact object regions: it first refines these coarse object tokens and only afterward refines the patch tokens, so that moderate imperfections in the initial clusters can be absorbed during refinement.
>
> Moreover, to mitigate typical failure modes of SSL-guided masks, we apply Voronoi clustering with farthest-point sampling to define attention masks, which enforces spatially diverse cluster centers. When DINO features focus on a single large object, this sampling tends to distribute centers across object and background rather than collapsing all prompts onto one region; when small or thin objects are slightly over-segmented, ORA’s multi-scale refinement and object–patch interactions help aggregate nearby regions with similar appearance. In addition, as detailed in our response to Weakness 1 and the new very-few-seen ablation (Table A), ORA together with SSL-guided prompts drives quantitative improvements in performance, and OC-ZSS does not require perfectly accurate SSL clusters as long as they provide reasonably meaningful object cues.
>
> Empirically also, OC-ZSS consistently improves over OTSeg on PASCAL Context and COCO-Stuff, which include small, thin, cluttered, and large objects (Tables 1 and 2). A per-class analysis on COCO-Stuff (reported in the appendix) shows that refinement brings substantial gains for several fine-grained or cluttered categories, while more ambiguous or near-saturated classes remain challenging:
>
> **Table B: Example per-class IoU on COCO-Stuff (OTSeg vs OC-ZSS)**
>
> | Class         | OTSeg IoU | OC-ZSS IoU | Δ IoU |
> | ------------- | --------- | ---------: | ----: |
> | wall-concrete | 17.96     |      30.75 | 12.79 |
> | skateboard    | 40.17     |      46.31 |  6.14 |
> | baseball bat  | 33.87     |      37.41 |  3.54 |
> | stairs        | 17.93     |      20.55 |  2.62 |
> | road          | 38.45     |      42.12 |  3.67 |
> | clouds        | 32.90     |      34.12 |  1.22 |
> | giraffe       | 68.17     |      68.85 |  0.68 |
> | elephant      | 87.84     |      87.95 |  0.11 |
>
> We agree that ORA cannot fully repair arbitrarily poor initial groupings and that a complete characterization of DINO’s biases is beyond the scope of this work. Nonetheless, the combination of a strong but generic SSL prior, a clustering scheme designed to avoid degenerate masks, and an object-aware refinement module offers a practical trade-off between robustness and flexibility, with the additional benefit that the SSL component is modular and can be replaced by future advances in self-supervised or unsupervised object discovery.

---

> ### Author Response · Authors · 2025-11-21
> **Review Response - 3/6**
>
> **Weakness 3**
>
> We thank the reviewer for raising this point. In our implementation, CLIP patch tokens are reshaped back to the regular ((H × W)) grid before applying the multi-scale module, so the atrous depthwise convolutions operate on **contiguous** spatial neighborhoods of ViT patch features. Although ViT uses non-overlapping patches, neighboring tokens still correspond to adjacent image regions, and the convolutions act as a local smoothing and aggregation operator over this grid, thereby reducing sharp discontinuities. We clarified this step more explicitly in the implementation details in the appendix.
>
> Our granular attention acts as a lightweight module that replaces the linear Q/K/V projections in ORA with a small multi-scale block composed of depthwise atrous convolutions followed by a ((1 × 1)) projection. In practice, we use dilation rates ((1, 6, 12, 18)) and implement the block with depthwise convolutions. This design yields consistent improvements: on VOC 2012, hIoU increases from 88.3 to 89.2 when adding granular attention on top of ORA with linear Q/K/V (Table 6), while keeping the refinement module computationally lightweight.
>
> Additionally, convolution-based multi-scale modules have been shown to work well in recent attention-based segmentation architectures [14], which supports our choice of a convolutional multi-scale layer within ORA. We agree that exploring transformer-native multi-scale mechanisms (e.g., hierarchical or windowed multi-scale attention) inside ORA is an interesting direction. We view such alternatives as orthogonal future work, while our current contribution focuses on object-centric refinement with a simple, efficient multi-scale adapter.
>
> **Weakness 4**
>
> We thank the reviewer for this observation and clarify that the Sinkhorn algorithm in cross-attention of the decoder is taken directly from OTSeg as our decoder is based on OTSeg. We do not modify any hyperparameters related to Sinkhorn attention (ε, number of iterations, log-domain stabilization), precisely to keep its behavior identical to OTSeg and isolate the effect of our object-centric refinement modules. Thus, OC-ZSS inherits the same stability and computational characteristics as OTSeg, while our contributions focus on improving CLIP patch representations rather than altering the optimal-transport decoder.
>
>
> **Weakness 5**
>
> We thank the reviewer for raising this point and clarify what we mean by “parameter-efficient” and “lightweight.” By parameter-efficient, we refer to the fact that **we do not fine-tune the CLIP vision or text encoders**: our trainable components (ORA and the granular multi-scale module) are small adapters, and the additional DINO encoder is kept **frozen**. As reported in the paper, OC-ZSS uses slightly more parameters than OTSeg with comparable GFLOPs, but adds fewer parameters and FLOPs than recent CLIP-based decoders such as CLIP-RC, which stacks multiple cross-attention layers in each decoder block and includes an additional recovery decoder. Moreover, OC-ZSS requires substantially fewer GFLOPs than two-stage networks like ZZSeg and ZegFormer that rely on a complex region-proposal stage. We also note that OC-ZSS is label-efficient, as shown in Table 5, where it improves hIoU from 29.4 to 64.8 in the 25% seen-class VOC setting (+35.4 hIoU over OTSeg).
>
> Regarding the specific points:
> **(a)** We agree that running a frozen DINO encoder increases inference cost and that this reliance is a limitation, and we leave designing object feature extraction that does not rely on an external SSL backbone to future work.
> **(b)** The Voronoi clustering used to obtain SSL-guided masks is implemented following the efficient PyTorch implementation in FPTrans [12], and in practice contributes only negligible overhead compared to the CLIP forward pass (Table 13).
> **(c)** The 4 dual-stage refinement iterations are designed to be cheaper than the decoder (Table 13): the Sinkhorn-based decoder operates on ((K × N)) prompt–class combinations and all patch tokens, whereas ORA operates on a small fixed number (6–8) of object prompts and patch tokens. Thus, the iterative refinement adds modest compute relative to the decoder while enabling the object-centric improvements that are the focus of this work.

---

> ### Author Response · Authors · 2025-11-21
> **Review Reponse - 4/6**
>
> **Weakness 8**
>
> We thank the reviewer for this comment. While OC-ZSS uses slightly more parameters than OTSeg (27.2M vs. 13.8M; Table 4), this increase is still moderate compared to other CLIP-based decoders such as CLIP-RC (36.9M parameters), and it is accompanied by consistent accuracy gains.
>
> In the **inductive setting**, where no unseen-class labels are available (Table 1), OC-ZSS improves unseen IoU over OTSeg on all three benchmarks. On VOC 2012, OC-ZSS achieves 85.3 unseen IoU vs. 81.6 for OTSeg; on PASCAL Context, 62.1 vs. 60.4; and on COCO-Stuff, 42.8 vs. 41.8. OC-ZSS also improves over the strong CLIP-RC baseline by approximately 4–5 points in seen, unseen IoU across datasets. Thus, even in the strict inductive regime, our OC-ZSS modules translate into consistent improvements and stronger generalization.
>
> In the **transductive setting** (Table 2), the gains are more pronounced, since the superior generalization of OC-ZSS is beneficial for generating unseen masks for self-training: OC-ZSS reaches 95.2 / 61.9 / 51.5 hIoU on VOC 2012, PASCAL Context, and COCO-Stuff, respectively, compared to 94.3 / 59.8 / 49.8 hIoU for OTSeg. The benefit is especially clear in the **very-few-seen regime**: in the VOC 25% setting (Table 5), OC-ZSS improves hIoU from 29.4 to 64.8 (+35.4 over OTSeg), indicating that the added capacity is effectively used to enhance generalization to unseen classes under limited supervision.
>
> Finally, we note that VOC, PASCAL Context, and COCO-Stuff are relatively saturated benchmarks on which strong classical segmentation architectures [14] already perform very well; on such datasets, absolute gains of 1–2 IoU are non-trivial. In this context, we view the moderate increase in parameters, together with consistent improvements in inductive, transductive, and very-few-seen settings, as a justified trade-off.
>
> **Weakness 6, 7, 9**
>
> We have updated the related work section to explicitly discuss SAM-CLIP [1], Delving into Shape-aware Zero-shot Semantic Segmentation [2], CLIP-DIY [3], Refining CLIP’s Spatial Awareness [4], and A Simple Framework for Open-Vocabulary Zero-Shot Segmentation [5], in addition to CLIP-DINOiser, ProxyCLIP, OVSegmentor, and FOSSIL (already covered in our original related work). However, these methods are **not directly comparable** to OC-ZSS due to substantial differences in training data, supervision, and evaluation protocols.
>
> Most of these works are designed for **open-vocabulary segmentation** [1–9] and deliberately move away from the standard zero-shot segmentation (ZSS) protocol used in CLIP-RC and OTSeg. Some, although named zero-shot segmentation, depart from this protocol: for example, [5] trains on COCO Captions and LAION-400M and then directly evaluates on datasets such as VOC, PASCAL Context, and COCO-Stuff without seen/unseen splits, while [4] fine-tunes the entire CLIP backbone with a new SCD framework on COCO 2017. [2] follows a few-shot-style setup, both in its training and evaluation on PASCAL-5(^i) and COCO-20(^i). Similarly, SAM-CLIP [1] requires training on large pre-training datasets such as CC3M, CC12M, and YFCC-15M and targets a different open-vocabulary protocol. In contrast, OC-ZSS operates in the **standard ZSS setting** with explicit seen/unseen splits and inductive/transductive evaluation on VOC 2012, PASCAL Context, and COCO-Stuff, using only seen-class masks and keeping CLIP frozen.
>
> Although these approaches are therefore not “apples-to-apples” baselines, we provide a coarse numerical comparison in the appendix (and below) by comparing our **transductive hIoU** with their reported **mIoU** on shared datasets where available.
>
> **Table C: Comparison with open vocabulary segmentation**
>
> | Method                                 | VOC mIoU | PASCAL Context mIoU | COCO-Stuff mIoU |
> |----------------------------------------|:--------:|:-------------------:|:---------------:|
> | SAM-CLIP                               | 60.6     | 29.2                | 31.5            |
> | CLIP-DIY                               | 59.9     | –                   | –               |
> | CLIP-DINOiser                          | ≈62.1    | ≈32.4               | ≈24.6           |
> | ProxyCLIP                              | 61.3     | 35.3                | 26.5            |
> | OVSegmentor                            | 53.8     | 20.4                | –               |
> | FOSSIL                                 | –        | 35.8                | 24.8            |
> | Simple Framework for OV Zero-Shot Seg. | 58.4     | 37.2                | –               |
> | OC-ZSS                                 | 95.2     | 61.9                | 51.5            |
>
>
> ---

---

> ### Author Response · Authors · 2025-11-21
> **Review Reponse - 5/6**
>
> **Minor 1, Q1, Q6**
>
> We provide ablations on VOC for both the number of object prompts (equal to the number of clusters) and the number of refinement iterations. The latest ablation for the number of object prompts is included in the appendix (Table 14), and the ablation for ORA iterations was already reported in Table 10. In both cases, the final values are chosen via a simple hyperparameter grid search on VOC: (S = 4, (n_o = 6)); COCO-Stuff: (S = 4, (n_o = 6)); PASCAL Context: (S = 5, (n_o = 8)). We do not perform any feature-layer selection for the SSL model, as our current setup using last-layer DINO patch features already works well in practice. The clustering itself follows the Voronoi + farthest-point sampling implementation of FPTrans [12], and introduces no learnable parameters or additional clustering hyperparameters beyond the number of clusters. Finally, the training iterations (20K / 40K / 80K for VOC / PASCAL Context / COCO-Stuff) follow OTSeg exactly, to ensure a fair comparison and avoid attributing gains to longer training rather than to the proposed refinement.
>
>
> **Minor 2**
>
>  We additionally evaluate on **ADE20K** by directly applying the COCO-Stuff trained checkpoints of OTSeg and OC-ZSS (without any retraining). As shown in Table D, OC-ZSS achieves higher mIoU than OTSeg (19.4 → 20.6), indicating that our object-centric refinement transfers well to a more diverse scene-parsing dataset.
>
>  **Table D: ADE20K mIoU with COCO-Stuff–trained models (no retraining).**
>
>  | Method | ADE20K mIoU |
>  | ------ | :---------: |
>  | OTSeg  |     19.4    |
>  | OC-ZSS |     20.6    |
>
> **Minor 3**
>
> Cross-dataset generalization (Table 3) is not our primary objective, and all methods show very small margins in this setting. Our focus is generalization to unseen classes which can be verified in main results and particularly under very limited supervision, where OC-ZSS shows large gains over OTSeg (e.g., on VOC with 25% seen classes, unseen mIoU/hIoU: 17.7/29.4 → 49.8/64.8; Table 5).
>
> **Minor 4 and 5**
>
> We follow common practice in semantic segmentation and set the dilation rates ({r_1, r_2, r_3, r_4} = {1, 6, 12, 18}), as used in DeepLab-style[15] multi-scale modules, and keep them **fixed** across datasets. We did not tune these rates further, as our goal is to obtain gains primarily from the proposed refinement design rather than from aggressive hyperparameter tweaking. The missing FOSSIL [9] reference at line 132 has been added in the revised manuscript.
>
> ---
>
> **Questions**
>
> **Q2 (Why dual-stage?).**
> SSL-guided object tokens are initially coarse, so we first refine *objects* and then use these refined objects to update *patches*. Replacing this dual-stage ORA with single-stage cross-attention or Slot Attention consistently lowers performance (Table 8, Table A).
>
> **Q3 (Why GRU?).**
> We use a GRU to stably update object tokens across ORA iterations; keeping the GRU but swapping ORA for standard cross-attention still degrades performance (Table 8, Table A), so the gains come from the GRU **plus** the dual-stage design, not from the GRU alone.
>
> **Q4**
> We use masked attention to **anchor each object prompt to a region** so prompts don’t all collapse onto the same dominant area. Global context is still carried by CLIP features and interactions between multiple prompts. Unmasked variants perform worse (Table 8), and our COCO-Stuff per-class analysis (Table B) shows clear gains on cluttered / fine-grained classes (e.g., wall-concrete, skateboard, baseball bat), indicating that masking improves object localization rather than hurting global context.
>
> **Q5 (What makes patches object-centric?).**
> We first capture **coarse object features** by grouping DINO patches into clusters (hence coarse object tokens), then refine patch features with these object features. The ablations in Table A show that this object-to-patch refinement gives a large boost over using patch refinement alone, indicating that patches become more object-aligned rather than just better clustered by texture.
>
> **Q7, Q8**
> We will release **full code and checkpoints upon acceptance**, including all hyperparameters and our Voronoi clustering implementation (following FPTrans [12]). This will make all implementation details directly inspectable. Concretely: (i) single GRU layer is shared across all iterations; (ii) the multi-scale atrous convolutions are applied **in parallel** and their outputs are fused; and (iii) we will clarify in the text how we construct masks from DINO clusters and use them to restrict the attention of object prompts and ORA.

---

> ### Author Response · Authors · 2025-11-21
> **Reviewe Reponse - 6/6**
>
> **References**
>
> [1] SAM-CLIP: Merging Vision Foundation Models towards Semantic and Spatial Understanding. CVPR Workshops, 2024.
>
> [2] Delving into Shape-aware Zero-shot Semantic Segmentation. CVPR, 2023.
>
> [3] CLIP-DIY: CLIP Dense Inference Yields Open-Vocabulary Semantic Segmentation For-Free. WACV, 2024.
>
> [4] Refining CLIP’s Spatial Awareness: A Visual-Centric Perspective. ICLR, 2025.
>
> [5] A Simple Framework for Open-Vocabulary Zero-Shot Segmentation. ICLR, 2025.
>
> [6] CLIP-DINOiser: Teaching CLIP a few DINO tricks for Open-Vocabulary Semantic Segmentation. ECCV, 2024.
>
> [7] ProxyCLIP: Proxy Attention Improves CLIP for Open-Vocabulary Semantic Segmentation. ECCV, 2024.
>
> [8] Learning Open-vocabulary Semantic Segmentation Models From Natural Language Supervision. CVPR, 2023.
>
> [9] FOSSIL: Free Open-Vocabulary Semantic Segmentation through Synthetic References Retrieval. WACV, 2024.
>
> [10] Learning Mask-aware CLIP Representations for Zero-shot Segmentation. NeurIPS, 2023.
>
> [12] Feature-Proxy Transformer for Few-Shot Segmentation. NeurIPS, 2022.
>
> [13] Unlocking Slot Attention by Changing Optimal Transport Costs. PMLR, 2023.
>
> [14] SegNeXt: Rethinking Convolutional Attention Design for Semantic Segmentation. NeurIPS, 2022.
>
> [15] Encoder-Decoder with Atrous Separable Convolution for Semantic Image Segmentation. ECCV, 2018

---

> ### Author Response · Authors · 2025-11-28
> **Gentle Remainder**
>
> A brief note to kindly follow up with the reviewers: we appreciate your detailed feedback, which helped us clarify and strengthen the paper. If there are any remaining concerns, we are happy to address them; if not, we would be grateful if you could consider increasing the score where you feel it is appropriate.

---

### Meta-Review · Area_Chair_gMUm · 2026-01-06

**Summary:**

In the initial phase, all reviewers recognised the contributions of the paper and gave positive scores. However, Some reviewers have concerns about novelty and stability of the proposed method. Some think the paper lacks proper computation comparisons, in-depth analysis, and additional experiments.

Reviewer 3xo5
1. The novelty of the paper is moderate. The paper lacks conceptual depth or theoretical analysis.
2. The proposed method relies on external models.
3. The use of taros convolutions within the granular attention mechanism may not be proper.
4. The use of entropy-regularized Sinkhorn attention leads concerns about training stability, computational efficiency, and generalization sensitivity.
5. Lack of additional experiments and comparisons.

Reviewer wdgB
1.  Improper presentations of memory and computation costs.
2.  The main performance gain might come from the strong semantics of the DINO encoder rather than other proposed components.
3. An ablation study to test where the few-seen gains come from is required.

Reviewer JToE
1. This proposed approach relies on external models. This concern is similar to 2nd concern of Reviewer 3xo5.
2.  Limited theoretical insight regarding the iterative ORA process.
3.  Fine-grained evaluations are required.
4. Additional comparisons with stronger backbones are required.

**Reviewer Concerns:**

For all concerns, the authors provided detained responses, including well-designed experiments and thorough analysis.

All concerns have been addressed.

**Reviewer Scores:**

In my opinion, after reading the rebuttals, some reviewers may increase their scores to 8 (clear accept) while others may keep their initial scores.

---

### Decision · Program_Chairs · 2026-01-26

Accept (Poster)